# SoftHebb: Bayesian inference in unsupervised Hebbian soft winner-take-all networks

## Abstract

State-of-the-art artificial neural networks (ANNs) require labelled data or feedback between layers, are often biologically implausible, and are vulnerable to adversarial attacks that humans are not susceptible to. On the other hand, Hebbian learning in winner-take-all (WTA) networks is unsupervised, feed-forward, and biologically plausible. However, a modern objective optimization theory for WTA networks has been missing, except under very limiting assumptions. Here we derive formally such a theory, based on biologically plausible but generic ANN elements. Through Hebbian learning, network parameters maintain a Bayesian generative model of the data. There is no supervisory loss function, but the network does minimize cross-entropy between its activations and the input distribution. The key is a "soft" WTA where there is no absolute "hard" winner neuron, and a specific type of Hebbian-like plasticity of weights and biases. We confirm our theory in practice, where, in handwritten digit (MNIST) recognition, our Hebbian algorithm, SoftHebb, minimizes cross-entropy without having access to it, and outperforms the more frequently used, hard-WTA-based method. Strikingly, it even outperforms supervised end-to-end backpropagation, under certain conditions. Specifically, in a two-layered network, SoftHebb outperforms backpropagation when the training dataset is only presented once, when the testing data is noisy, and under gradient-based adversarial attacks. Notably, adversarial attacks that maximally confuse SoftHebb are also confusing to the human eye. Finally, the model can generate interpolations of objects from its input distribution. All in all, SoftHebb extends Hebbian WTA theory with modern machine learning tools, thus making these networks relevant to pertinent issues in deep learning.

## 1 Introduction

State-of-the-art (SOTA) artificial neural networks (ANNs) achieve impressive results in a variety of machine intelligence tasks (Sejnowski, 2020). However, they largely rely on mechanisms that diverge from the original inspiration from biological neural networks (Bengio et al., 2015; Illing et al., 2019). As a result, only a small part of this prolific field also contributes to computational neuroscience. In fact, this biological implausibility is also an important issue for machine intelligence. For their impressive performance, ANNs trade off other desired properties, which are present in biological systems. For example, ANN training often demands very large and labelled datasets. When labels are unavailable, self-supervised learning schemes exist, where supervisory error signals generated by the network itself are exploited and backpropagated from the output towards the input to update the network's parameters (Goodfellow et al., 2014; Devlin et al., 2018; Chen et al., 2020). However, this global propagation of signals in deep networks introduces another limitation. Namely, it prevents the implementation of efficient distributed computing hardware that would be based on only local signals from neighbouring physical nodes in the network, and is in contrast to local synaptic plasticity rules that partly govern biological learning. Several pieces of work have been addressing parts of the biological implausibility and hardware-inefficiency of backpropagation in ANNs (Bengio et al., 2015; Lillicrap et al., 2016; Guerguiev et al., 2017; Pfeiffer & Pfeil, 2018; Illing et al., 2019; Pogodin & Latham, 2020; Millidge et al., 2020; Pogodin et al., 2021). such as the need for exactly symmetric forward and backward weights or the waiting time caused by the network's forward-backward pass between two training updates in a layer (weight transport and update-locking problems). Recently, an approximation to backpropagation that is mostly Hebbian, i.e. relies on mostly pre- and post-

synaptic activity of each synapse, has been achieved by reducing the global error requirements to 1-bit information (Pogodin & Latham, 2020). Two schemes that further localize the signal that is required for a weight update are Equilibrium Propagation (Scellier & Bengio, 2017) and Predictive Coding (Millidge et al., 2020). Both methods approximate backpropagation through Hebbian-like learning, by delegating the global aspect of the computation, from a global error signal, to a global convergence of the network state to an equilibrium. This equilibrium is reached through several iterative steps of feed-forward and feed-back communication throughout the network, before the ultimate weight update by one training example. The biological plausibility and hardware-efficiency of this added iterative process of signal propagation are open questions that begin to be addressed (Ernoult et al., 2020).

Moreover, learning through backpropagation, and presumably also its approximations, has another indication of biological implausibility, which also significantly limits ANN applicability. Namely, it produces networks that are confused by small adversarial perturbations of the input, which are imperceptible by humans. It has recently been proposed that a defence strategy of "deflection" of adversarial attacks may be the ultimate solution to that problem (Qin et al., 2020). Through this strategy, to cause confusion in the network's inferred class, the adversary is forced to generate such a changed input that really belongs to the distribution of a different input class. Intuitively, but also strictly by definition, this deflection is achieved if a human assigns to the perturbed input the same label that the network does. Deflection of adversarial attacks in ANNs has been demonstrated by an elaborate scheme that is based on detecting the attacks (Qin et al., 2020). However, the human ability to deflect adversarial perturbations likely does not rely on detecting them, but rather on effectively ignoring them, making the deflecting type of robustness an emergent property of biological computation rather than a defence mechanism. The biological principles that underlie this property of robustness are unclear, but it might emerge from the distinct algorithms that govern learning in the brain.

Therefore, what is missing is a biologically plausible model that can learn from fewer data-points, without labels, through local plasticity, and without feedback from distant layers. This model could then be tested for emergent adversarial robustness. A good candidate category of biological networks and learning algorithms is that of competitive learning. Neurons that compete for their activation through lateral inhibition are a common connectivity pattern in the superficial layers of the cerebral cortex (Douglas & Martin, 2004; Binzegger et al., 2004). This pattern is described as winner-take-all (WTA), because competition suppresses activity of weakly activated neurons, and emphasizes strong ones. Combined with Hebbian-like plasticity rules, WTA connectivity gives rise to competitive-learning algorithms. These networks and learning schemes have been long studied (Von der Malsburg, 1973) and a large literature based on simulations and analyses describes their functional properties. A WTA neuronal layer, depending on its specifics, can restore missing input signals (Rutishauser et al., 2011; Diehl & Cook, 2016), perform decision making i.e. winner selection (Hahnloser et al., 1999; Maass, 2000; Rutishauser et al., 2011), and generate oscillations such as those that underlie brain rhythms (Cannon et al., 2014). Perhaps more importantly, its neurons can learn to become selective to different input patterns, such as orientation of visual bars in models of the primary visual cortex (Von der Malsburg, 1973), MNIST handwritten digits (Nessler et al., 2013; Diehl & Cook, 2015; Krotov & Hopfield, 2019), CIFAR10 objects (Krotov & Hopfield, 2019), spatiotemporal spiking patterns (Nessler et al., 2013), and can adapt dynamically to model changing objects (Moraitis et al., 2020). The WTA model is indeed biologically plausible, Hebbian plasticity is local, and learning is input-driven, relying on only feed-forward communication of neurons – properties that seem to address several of the limitations of ANNs. However, the model's applicability is limited to simple tasks. That is partly because the related theoretical literature remains surprisingly unsettled, despite its long history, and the strong and productive community interest (Sanger, 1989; Földiák & Fdilr, 1989; Földiak, 1990; Linsker, 1992; Olshausen & Field, 1996; Bell & Sejnowski, 1995; Olshausen & Field, 1997; Lee et al., 1999; Nessler et al., 2013; Pehlevan & Chklovskii, 2014; Hu et al., 2014; Pehlevan & Chklovskii, 2015; Pehlevan et al., 2017; Isomura & Toyoizumi, 2018).

Nessler et al. (2009; 2013) described a very related theory but for a model that is largely incompatible with ANNs and thus less practical. It uses spiking and stochastic neurons, input has to be discretized, and each input feature must be encoded through multiple binary neurons. Moreover, it was only proven for neurons with an exponential activation function. It remains therefore unclear which specific plasticity rule and structure could optimize an ANN WTA for Bayesian inference. It is also unclear how to minimize a common loss function such as cross-entropy despite unsupervised learning,

and how a WTA could represent varying families of probability distributions. In summary, on the theoretical side, an algorithm that is simultaneously normative, based on WTA networks and Hebbian unsupervised plasticity, performs Bayesian inference, and, importantly, is composed of conventional, i.e. non-spiking, ANN elements and is rigorously linked to modern ANN tools such as cross-entropy loss, would be an important advance but has been missing. On the practical side, evidence that Hebbian WTA networks could be useful for presently pertinent issues of modern ANNs such as adversarial robustness, generation of synthetic images, or faster learning, has remained limited. Here we aim to fill these gaps. Recently, when WTA networks were studied in a theoretical framework compatible with conventional machine learning (ML), but in the context of short-term as opposed to long-term Hebbian plasticity, it resulted in surprising practical advantages over supervised ANNs (Moraitis et al., 2020). A similar theoretical approach could also reveal unknown advantages of long-term Hebbian plasticity in WTA networks. In addition, it could provide insights into how a WTA microcircuit could participate in larger-scale computation by deeper cortical or artificial networks.

Here we construct "SoftHebb", a biologically plausible WTA model that is based on standard rate-based neurons as in ANNs, can accommodate various activation functions, and learns without labels, using local plasticity and only feed-forward communication, i.e. the properties we seek in an ANN. Importantly, it is equipped with a simple normalization of the layer's activations, and an optional temperature-scaling mechanism (Hinton et al., 2015), producing a soft WTA instead of selecting a single "hard" winner neuron. This allows us to prove formally that a SoftHebb layer is a generative mixture model that objectively minimizes its Kullback-Leibler (KL) divergence from the input distribution through Bayesian inference, thus providing a new formal ML-theoretic perspective of these networks. We complement our main results, which are theoretical, with experiments that are small-scale but produce intriguing results. As a generative model, SoftHebb has a broader scope than classification, but we test it on image classification tasks. Surprisingly, in addition to overcoming several inefficiencies of backpropagation, the unsupervised WTA model also outperforms a supervised two-layer perceptron in several aspects: learning speed and accuracy in the first presentation of the training dataset, robustness to noisy data and to one of the strongest white-box adversarial attacks, i.e. projected gradient descent (PGD) (Madry et al., 2017), and without any explicit defence. Interestingly, the SoftHebb model also exhibits inherent properties of deflection (Qin et al., 2020) of the adversarial attacks, and generates object interpolations.

## 2 THEORY

A supporting diagram summarising the theoretical and neural model, and a succinct description of the learning algorithm are provided in the beginning of Appendix A.

**Definition 2.1 (The input assumptions).** *Each observation $_j\boldsymbol{x} \in \mathbb{R}^n$ is generated by a hidden "cause" $_jC$ from a finite set of $K$ possible such causes: $_jC \in \{C_k, \forall k \leq K \in \mathbb{N}\}$. Therefore, the data is generated by a mixture of the probability distributions attributed to each of the $K$ classes $C_k$:*

$$p(\boldsymbol{x}) = \sum_{k=1}^{K} p(\boldsymbol{x}|C_k)P(C_k). \tag{1}$$

*In addition, the dimensions of $\boldsymbol{x}$, $x_i$ are conditionally independent from each other, i.e.*
*$p(\boldsymbol{x}) = \prod_{i=1}^{n} p(x_i)$. The number $K$ of the true causes or classes of the data is assumed to be known.*

The term "cause" is used here in the sense of causal inference. It is important to emphasize that the true cause of each input is hidden, i.e. not known. In the case of a labelled dataset, labels may correspond to causes, and the labels are deleted before presenting the training data to the model. We choose a mixture model that corresponds to the data assumptions but is also interpretable in neural terms (Paragraph 2.4):

**Definition 2.2 (The generative probabilistic mixture model).** *We consider a mixture model distribution $q$: $q(\boldsymbol{x}) = \sum_{k=1}^{K} q(\boldsymbol{x}|C_k) Q(C_k)$, approximating the data distribution $p$. We choose specifically a mixture of exponentials and we parametrize $Q(C_k; w_{0k})$ also as an exponential, specifically:*

$$q(x_i|C_k; w_{ik}) = e^{w_{ik} \cdot \frac{x_i}{||\boldsymbol{x}||}}, \forall k \tag{2}$$

$$Q(C_k; w_{0k}) = e^{w_{0k}}, \forall k. \tag{3}$$

*In addition, the parameter vectors are subject to the normalization constraints: $||\boldsymbol{w}_k|| = 1$, $\forall k$, and $\sum_{k=1}^{K} e^{w_{0k}} = 1$.*

The model we have chosen is a reasonable choice because it factorizes similarly to the data of Definition 2.1:

$$q_k := q(\boldsymbol{x}|C_k; \boldsymbol{w}_k) = \prod_{i=1}^{n} q(x_i|C_k; w_{ik}) = e^{\sum_{i=1}^{n} w_{ik} \frac{x_i}{||\boldsymbol{x}||}} = e^{u_k}, \qquad (4)$$

where $u_k = \frac{\boldsymbol{w}_k \cdot \boldsymbol{x}}{||\boldsymbol{w}_k|| \cdot ||\boldsymbol{x}||}$, i.e. the cosine similarity of the two vectors. A similar probabilistic model was used in related previous theoretical work Nessler et al. (2009; 2013); Moraitis et al. (2020), but for different data assumptions, and with certain further constraints to the model. Namely, (Nessler et al., 2009; 2013) considered data that was binary, and created by a population code, while the model was stochastic. These works provide the foundation of our derivation, but here we consider the more generic scenario where data are continuous-valued and input directly into the model, which is deterministic and, as we will show, more compatible with standard ANNs. In Moraitis et al. (2020), data had particular short-term temporal dependencies, whereas here we consider the distinct case of independent and identically distributed (i.i.d.) input samples. The Bayes-optimal parameters of a model mixture of exponentials can be found analytically as functions of the input distribution's parameters, and the model is equivalent to a soft winner-take-all neural network (Moraitis et al., 2020). After describing this, we will prove here that Hebbian plasticity of synapses combined with local plasticity of the neuronal biases sets the parameters to their optimal values.

**Theorem 2.3 (The optimal parameters of the model).** *The parameters that minimize the KL divergence of such a mixture model from the data are, for every $k$,*

$$_{opt}w_{0k} = \ln P(C_k) \qquad (5)$$

$$and \ _{opt}\boldsymbol{w}_k^* = \frac{_{opt}\boldsymbol{w}_k}{||\ _{opt}\boldsymbol{w}_k||} = \frac{\mu_{p_k}(\boldsymbol{x})}{||\mu_{p_k}(\boldsymbol{x})||}, \qquad (6)$$

*where $_{opt}\boldsymbol{w}_k = c \cdot \mu_{p_k}(\boldsymbol{x}), c \in \mathbb{R}^+$, $\mu_{p_k}(\boldsymbol{x})$ is the mean of the distribution $p_k$, and $p_k := p(\boldsymbol{x}|C_k)$.*

In other words, the optimal parameter vector of each component $k$ in this mixture is proportional to the mean of the corresponding component of the input distribution, i.e. it is a centroid of the component. In addition, the optimal parameter of the model's prior $Q(C_k)$ is the logarithm of the corresponding component's prior probability. This Theorem's proof was provided in the supplementary material of Moraitis et al. (2020), but for completeness we also provide it in our Appendix. These centroids and priors of the input's component distributions, as well as the method of their estimation, however, are different for different input assumptions, and we will derive a learning rule that provably sets the parameters to their Maximum Likelihood Estimate for the inputs addressed here. The learning rule is a Hebbian type of synaptic plasticity combined with a plasticity for neuronal biases. Before providing the rule and the related proof, we will describe how our mixture model is equivalent to a WTA neural network.

### 2.4 EQUIVALENCE OF THE PROBABILISTIC MODEL TO A WTA NEURAL NETWORK

The cosine similarity between the input vector and each centroid's parameters underpins the model (Eq. 4). This similarity is precisely computed by a linear neuron that receives normalized inputs $\boldsymbol{x}^* := \frac{\boldsymbol{x}}{||\boldsymbol{x}||}$, and that normalizes its vector of synaptic weights: $\boldsymbol{w}_k^* := \frac{\boldsymbol{w}}{||\boldsymbol{w}||}$. Specifically, the neuron's summed weighted input $u_k = \boldsymbol{w}_k^* \cdot \boldsymbol{x}^*$ then determines the cosine similarity of an input sample to the weight vector, thus computing the likelihood function of each component of the input mixture (Eq. 2). It should be noted that even though $u_k$ depends on the weights of all input synapses, the weight values of other synapses do not need to be known to each updated synapse. Therefore, in the SoftHebb plasticity rule that we will present (Eq. 8), the term $u_k$ is a local, postsynaptic variable that does not undermine the locality of the plasticity. The bias term of each neuron can store the parameter $w_{0k}$ of the prior $Q(C_k; w_{0k})$. Based on these, it can also be shown that a set of $K$ such neurons can actually compute the Bayesian posterior, if the neurons are connected in a configuration that implements softmax. Softmax has a biologically-plausible implementation through lateral inhibition (divisive normalization) between neurons (Nessler et al., 2009; 2013; Moraitis et al., 2020). Specifically, based on the model of Definition 2.2, the posterior probability is

$$Q(C_k|\boldsymbol{x}) = \frac{e^{u_k + w_{0k}}}{\sum_{l=1}^{K} e^{u_l + w_{0l}}}. \qquad (7)$$

But in the neural description, $u_k + w_{0k}$ is the activation of the $k$-th linear neuron. That is, Eq. 7 shows that the result of Bayesian inference of the hidden cause from the input $Q(C_k|\boldsymbol{x})$ is found

by a softmax operation on the linear neural activations. In this equivalence, we will be using $y_k := Q(C_k|\boldsymbol{x}; \boldsymbol{w})$ to symbolize the softmax output of the $k$-th neuron, i.e. the output after the WTA operation, interchangeably with $Q(C_k|\boldsymbol{x})$. It can be seen in Eq. 7 that the probabilistic model has one more, alternative, but equivalent neural interpretation. Specifically, $Q(C_k|\boldsymbol{x})$ can be described as the output of a neuron with exponential activation function (numerator in Eq. 7) that is normalized by its layer's total output (denominator). This is equally accurate, and more directly analogous to the biological description (Nessler et al., 2009; 2013; Moraitis et al., 2020). This shows that the exponential activation of each individual neuron $k$ directly equals the $k$-th exponential component distribution of the generative mixture model (Eq. 4). Therefore, the softmax-configured linear neurons, or equivalently, the normalized exponential neurons, fully implement the generative model of Definition 2.2, and also infer the Bayesian posterior probability given an input and the model parameters. However, the problem of calculating the model's parameters from data samples is a difficult one, if the input distribution's parameters are unknown. In the next sections we will show that this neural network can find these optimal parameters through Bayesian inference, in an unsupervised and on-line manner, based on only local Hebbian plasticity.

## 2.5 A HEBBIAN RULE THAT OPTIMIZES THE WEIGHTS

Several Hebbian-like rules exist and have been combined with WTA networks. For example, in the case of stochastic binary neurons and binary population-coded inputs, it has been shown that weight updates with an exponential weight-dependence find the optimal weights (Nessler et al., 2009; 2013). Oja's rule is another candidate (Oja, 1982). An individual linear neuron equipped with this learning rule finds the first principal component of the input data (Oja, 1982). A variation of Oja's rule combined with hard-WTA networks and additional mechanisms has achieved good experimental results performance on classification tasks (Krotov & Hopfield, 2019), but lacks the theoretical underpinning that we aim for. Here we propose a Hebbian-like rule for which we will show it optimizes the soft WTA's generative model. The rule is similar to Oja's rule, but considers, for each neuron $k$, both its linear weighted summation of the inputs $u_k$, and its nonlinear output of the WTA $y_k$:

$$\Delta w_{ik}^{(SoftHebb)} := \eta \cdot y_k \cdot (x_i - u_k w_{ik}), \tag{8}$$

where $w_{ik}$ is the synaptic weight from the $i$-th input to the $k$-th neuron, and $\eta$ is the learning rate hyperparameter. As can be seen, all involved variables are local to the synapse, i.e. only indices $i$ and $k$ are relevant. No signals from distant layers, from non-perisynaptic neurons, or from other synapses are involved. By solving the equation $E[\Delta w_{ik}] = 0$ where $E[\cdot]$ is the expected value over the input distribution, we can show that, with this rule, there exists a stable equilibrium value of the weights, and this equilibrium value is an optimal value according to Theorem 2.3:

**Theorem 2.5.** *The equilibrium weights of the SoftHebb synaptic plasticity rule are*

$$w_{ik}^{SoftHebb} = c \cdot \mu_{p_k}(x_i) = {}_{opt}w_{ik}, \text{ where } c = \frac{1}{||\mu_{p_k}(\boldsymbol{x})||}. \tag{9}$$

The proof is provided in the Appendix. Therefore, our update rule (Eq. 8) optimizes the weights of the neural network.

## 2.6 LOCAL LEARNING OF NEURONAL BIASES AS BAYESIAN PRIORS

For the complete optimization of the model, the neuronal biases $w_{0k}$ must also be optimized to satisfy Eq. 5, i.e. to optimize the Bayesian prior belief for the probability distribution over the $K$ input causes. We define the following rate-based rule inspired from the spike-based bias rule of (Nessler et al., 2013):

$$\Delta w_{0k}^{SoftHebb} = \eta e^{-w_{0k}} \left(y_k - e^{w_{0k}}\right). \tag{10}$$

With the same technique we used for Theorem 2.5, we also provide proof in the Appendix that the equilibrium of the bias with this rule matches the optimal value ${}_{opt}w_{0k} = \ln P(C_k)$ of Theorem 2.3:

**Theorem 2.6.** *The equilibrium biases of the SoftHebb bias learning rule are*

$$w_{0k}^{SoftHebb} = \ln P(C_k) = {}_{opt}w_{0k}. \tag{11}$$

## 2.7 ALTERNATE ACTIVATION FUNCTIONS AND RELATION TO HARD WTA

The model of Definition 2.2 uses for each component $p(\boldsymbol{x}|C_k)$ an exponential probability distribution with a base of Euler's e, equivalent to a model using similarly exponential neurons (Subsection 2.4).

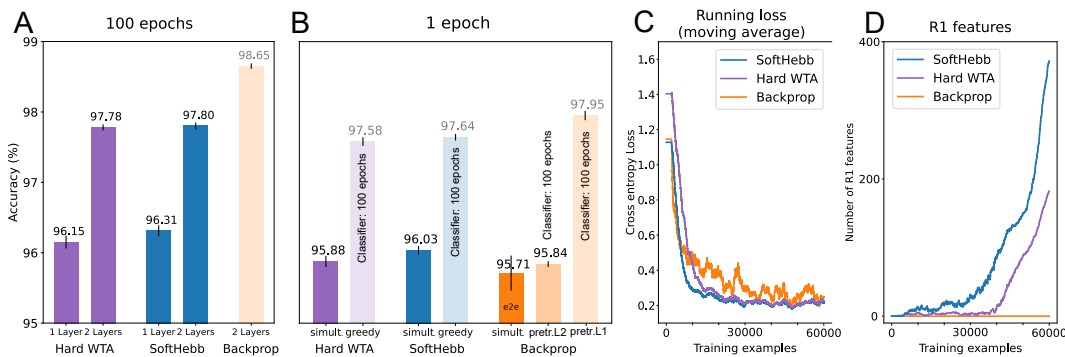

Figure 1: Performance of SoftHebb on MNIST compared to hard-WTA and backpropagation.

Depending on the task, different probability distribution shapes, i.e. different neuronal activation functions, may be better models. This is compatible with our theory (see Appendix B). Firstly, the base of the exponential activation function can be chosen differently, resulting in a softmax function with a different base, such that Eq. 7 becomes more generally $Q(C_k|\boldsymbol{x}) = \frac{b^{u_k+w_{0k}}}{\sum_{l=1}^{K} b^{u_l+w_{0l}}}$. This is equivalent to Temperature Scaling (Hinton et al., 2015), a mechanism that also maintains the probabilistic interpretation of the output. It can also be implemented by a normalized layer of exponential neurons, and are compatible with our theoretical derivations and the optimization by the plasticity rule of Eq. 8. This allows us to integrate the hard WTA into the SoftHebb framework, as a special case with an infinite base. Therefore, the hard WTA, if used with the plasticity rule that we derived, is expected to have some similar properties to a soft WTA implementation. Moreover, we show in the Appendix that soft WTA models can be constructed by rectified linear units (ReLU) or in general by neurons with any non-negative monotonically increasing activation function, and their weights are also optimized by the same plasticity rule.

### 2.8 POST-HOC CROSS-ENTROPY

It is not obvious how to measure the loss of the WTA network during the learning process, since the ground truth for causes $C$ may not match the labels, or even their number. For example, in reality, the cause $C$ of each MNIST example in the sense implied by causal inference is not the digit cause itself, but a combination of a single digit cause $D$, which is the MNIST label, with one of many handwriting styles $S$. In the Appendix C, we derive a method for measuring the loss minimization during training in spite of this mismatch. The method's steps can be summarised as (a) unsupervised training of SoftHebb, then (b) training a supervised classifier on top, and finally (c1) repeating the training of SoftHebb with the same initial weights and ordering of the training inputs, while (c2) measuring the trained classifier's loss. In this way, we can observe the cross-entropy loss $H^{labels}$ of SoftHebb while it is being minimized, and infer that $H^{causes}$ is also minimized (Eq. 72). We call this the post-hoc cross-entropy method, and we have used it in our experiments (Section 3.2 and Fig. 1 C and D) to evaluate the learning process in a theoretically sound manner.

## 3 EXPERIMENTS

### 3.1 MNIST TOP-ACCURACY BENCHMARKS

We implemented the theoretical SoftHebb model in simulations and tested it in the task of learning to classify MNIST handwritten digits. The network received the MNIST frames normalized by their Euclidean norm, while the plasticity rule that we derived updated its weights and biases in an unsupervised manner. We used $K = 2000$ neurons. First we trained the network for 100 epochs, i.e. randomly ordered presentations of the 60000 training digits. Each training experiment was repeated five times with varying random initializations and input order. We will report the mean and standard deviation of accuracies. Inference of the input labels by the WTA network of 2000 neurons was performed in two different ways. The first approach is single-layer, where, after training the network, we assigned a label to each of the 2000 neurons, in a standard approach that is used in unsupervised clustering. Namely, for each neuron, we found the label of the training set that makes it win the WTA

competition most often. In this single-layer approach, this is the only time when labels were used, and at no point were weights updated using labels. The second approach was two-layer and based on supervised training of a perceptron classifier on top of the WTA layer. The classifier layer was trained with the Adam optimizer and cross-entropy loss for 100 epochs, while the previously-trained WTA parameters were frozen. SoftHebb achieved an accuracy of $(96.31 \pm 0.06)\%$ and $(97.80 \pm 0.02)\%$ in its 1- and 2-layer form respectively. To test the strengths of the soft-WTA approach combined with training the priors through biases, which makes the network Bayesian, we also trained the weights of a hard-WTA network. The SoftHebb model slightly outperformed the hard WTA (Fig. 1A), especially in the 1-layer case where the supervised 2nd layer cannot compensate the drop. However, SoftHebb's accuracy is significantly lower than a multi-layer perceptron (MLP) with one hidden layer of also 2000 neurons that is trained exhaustively $((98.65 \pm 0.06)\%)$. This is not surprising, due to end-to-end training, supervision, and the MLP being a discriminative model as opposed to a generative model merely applied to a classification task, as SoftHebb is. If the Bayesian and generative aspects that follow from our theory were not required, several mechanisms exist to enhance the discriminative power of WTA networks (Krotov & Hopfield, 2019), and even a random projection layer instead of a trained WTA performs well (Illing et al., 2019). The generative approach however has its own advantages even for a discriminative task, and we will show some of these here.

## 3.2 Cross-entropy minimization. Speed advantage of SoftHebb.

Next, we tested SoftHebb's speed and efficiency by comparing it to other models during the first training epoch. In the common, "greedy" training of such networks, layer L+1 is trained only after layer L is trained on the full dataset and its weights frozen. We trained in this manner a second layer as a supervised classifier for 100 epochs. SoftHebb again slightly outperformed the Hard WTA showing that it extracts superior features. However, it was outperformed by 1-epoch backpropagation pretraining that was followed by 100 epochs of training the 2nd layer only (Fig. 1B, light-coloured bars). Then, we demonstrated that in a truly single-epoch scenario, without longer training for the second layer, SoftHebb further outperforms HardWTA, and, strikingly, it even outperforms end-to-end (e2e) backpropagation in accuracy (Fig. 1B, bars label "simult."). In fact, it even outperforms a backprop-trained network that is pretrained for 100 epochs, before its 1st layer is reset and re-trained end-to-end for 1 epoch (Fig. 1B, "pretr. L2"). In this experiment, the Hebbian networks had an additional important advantage. By using the delta rule for the 2nd layer, each individual training example updated both layers. In contrast to backpropagation, this simultaneous method does not suffer from the update-locking problem, i.e. the first layer can learn from the next example before the current input is even processed by the higher layer, let alone backpropagated. Moreover, we measured the post-hoc loss. Consistently with the first-epoch accuracy results, SoftHebb is faster than hard WTA and both are faster than backpropagation, which is remarkable considering the absence of input labels. Also, it validates our theory that SoftHebb minimizes cross-entropy. As a further insight into the differences between SoftHebb and hard-WTA learning, we measured throughout learning the number of learned features that lie on a hypersphere with a radius of $1 \pm 0.01$ (R1 features), according to the Euclidean norm of the weight vectors. The SoftHebb learning algorithm converges to such a normalization in theory (end of Appendix A, Theorem A.2) , and Fig. 1D validates that it does, but also that it does so faster than the hard WTA implementation, demonstrating SoftHebb's superiority in unsupervised representation learning, irrespective of discriminative ability. Hebbian learning compared to backpropagation has not generally been considered superior for its accuracy, but for other potential benefits. Here we show evidence that, for small problems demanding fast learning, SoftHebb may be superior to backpropagation even in terms of accuracy, in addition to its biological plausibility and efficiency.

## 3.3 Robustness to noise and adversarial attacks - Generative adv. properties

Based on the Bayesian, generative, and purely input-driven learning nature of the algorithm, we hypothesized that SoftHebb may be more robust to input perturbations. Indeed, we tested the trained SoftHebb and MLP models for robustness, and found that SoftHebb is significantly more robust than the backprop-trained MLP, both to added Gaussian noise and to PGD adversarial attacks (see Fig. 2). PGD (Madry et al., 2017) produces perturbations in a direction that maximizes the loss of each targeted network, and in size controlled by a parameter $\epsilon$. We also attacked a model where the Hebbian layer was trained as hard WTA but in forward propagation to the supervised classifier head and for inference, softmax as in SoftHebb was used. Hard- and soft-trained WTA are almost identically robust to these perturbations. This may be an expression of the fact that the hard-WTA model is essentially a special case of the SoftHebb framework (Section 2.7). Strikingly, the Hebbian

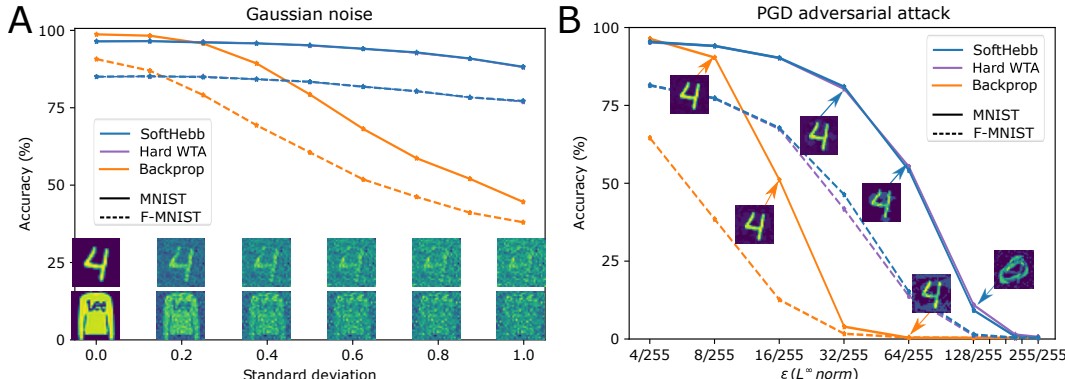

Figure 2: Noise and adversarial attack robustness of SoftHebb and of backpropagation-trained MLP on MNIST and Fashion-MNIST. The insets show one example from the testing set and its perturbed versions, for increasing perturbations. (A) SoftHebb is highly robust to noise. (B) MLP's MNIST accuracy drops to ~60% by hardly perceptible perturbations ($\epsilon = 16/255$), while SoftHebb requires visually noticeable perturbations ($\epsilon = 64/255$) for similar drop in performance. At that degree of perturbation, the MLP has already dropped to zero. SoftHebb deflects the attack: it forces the attacker to produce examples of truly different classes - the original digit "4" is perturbed to look like a "0" (see also Fig. 3). The hard-WTA curves (purple) are almost identical to SoftHebb's.

WTA model has a visible tendency to deflect the attacks, i.e. its most confusing examples actually belong to a perceptually different class (Fig. 2B and 3). This effectively nullifies the attack and was previously shown in elaborate SOTA adversarial-defence models (Qin et al., 2020). The attack's parameters were tuned systematically (Appendix E). The pair of the adversarial attacker with the generative SoftHebb model essentially composes a generative adversarial network (GAN), even though the term is usually reserved for pairs *trained* in tandem (Goodfellow et al., 2014; Creswell et al., 2018). As a result, the model could inherit certain properties of GANs. It can be seen that it is able to generate interpolations between input classes (Fig. 3). The parameter $\epsilon$ of the adversarial attack can control the balance between the interpolated objects. Similar functionality has existed in the realm of GANs (Radford et al., 2015), autoencoders (Berthelot et al., 2018), and other deep neural networks (Bojanowski et al., 2017), but was not known for simple biologically-plausible models.

### 3.4  EXTENSIBILITY OF SOFTHEBB: F-MNIST, CIFAR-10, CONV-SOFTHEBB

Finally, we performed preliminary tests on two more difficult datasets, namely Fashion-MNIST (Xiao et al., 2017), which contains grey-scale images of clothing products, and CIFAR-10 (Krizhevsky et al., 2009), which contains RGB images of animals and vehicles. We did not tune the Hebbian networks' hyper-parameters extensively, so accuracies on these tasks are not definitive but do give a good indication. On F-MNIST, the SoftHebb model achieved a top accuracy of $87.46\%$ whereas a hard WTA reached a similar accuracy of $87.49\%$. A supervised MLP of the same size achieved a test accuracy of $90.55\%$. SoftHebb's generative interpolations (Fig. 3B) are reconfirmed on the F-MNIST dataset, as is its robustness to attacks, whereas, with very small adversarial perturbations, the MLP drops to an accuracy lower than the SoftHebb model (dashed lines in Fig. 2). On CIFAR-10's preliminary results, the hard WTA and SoftHebb achieved an accuracy of $49.78\%$ and $50.27\%$ respectively. In every tested dataset, it became clear that SoftHebb learns faster than either backpropagation or a hard WTA, by observing the loss and the learned features as in Fig. 1 C & D. The fully-connected SoftHebb layer is applicable on MNIST because the data classes are well-clustered directly in the feature-space of pixels. That is, SoftHebb's probabilistic model's assumptions (Def. 2.1) are quite valid for this feature space, and increasing the number of neurons for discovering more refined sub-clusters does help. However, for more complex datasets, this approach alone has diminishing returns and multilayer networks will be needed. Towards this, we implemented a first version of a convolutional SoftHebb, with an added supervised classifier. In these early results, conv-SoftHebb achieved $98.63\%$ on MNIST, and $60.30\%$ on CIFAR-10. This confirms that conv-SoftHebb is functional, but we did not systematically optimize its accuracy or explore its other strengths. This feasibility is evidence that SoftHebb's strengths may soon also apply to larger networks and datasets.

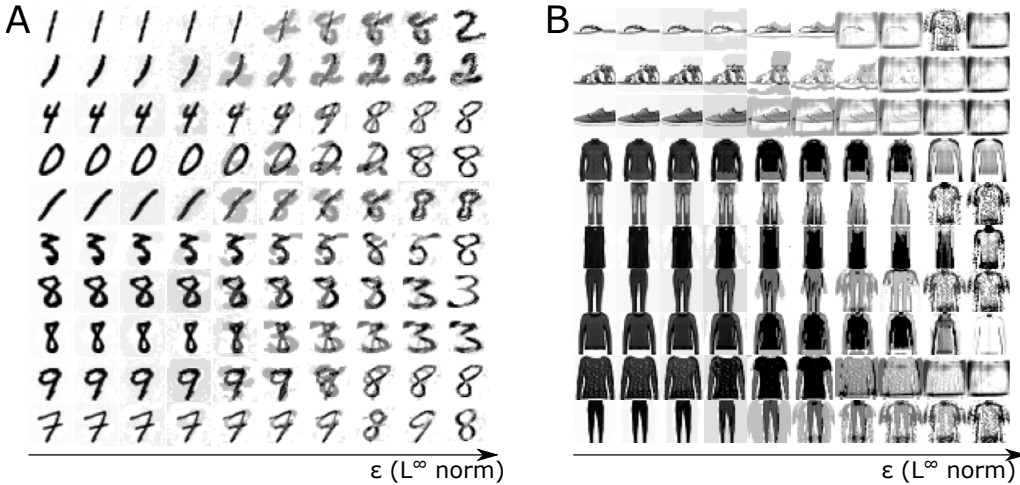

Figure 3: Examples generated by the adversarial pair PGD attacker/SoftHebb model. SoftHebb's inherent tendency to deflect the attack towards truly different classes is visible. This tendency can be repurposed to generate interpolations between different classes of the data distribution, a generative property previously unknown for such simple networks.

## 4 DISCUSSION

We have described SoftHebb, a highly biologically plausible neural algorithm that is founded on a Bayesian ML-theoretic framework. The model consists of elements fully compatible with conventional ANNs. It was previously not known which plasticity rule should be used to learn a Bayesian generative model of the input distribution in ANN WTA networks. Moreover, we showed that Hard WTA networks and neurons with other activation functions can be described within the same framework as variations of the probabilistic model. This theory could provide a new foundation for normative Hebbian ANN designs with practical significance. For example, SoftHebb's properties are sought-after by efficient neuromorphic learning chips. It is unsupervised, local, and requires no error or other feedback currents from upper layers, thus solving hardware-inefficiencies and bio-implausibilities of backpropagation such as weight-transport and update-locking. Surprisingly, it surpasses backpropagation even in accuracy, when training time and network size are limited. In a demonstration that goes beyond the common greedy-training approach to such networks, we achieved update-unlocked operation in practice, by updating the first layer before the input's full processing by the next layer. It is intriguing that, through its biological plausibility, emerge properties commonly associated with biological intelligence, such as speed of learning, and robustness to noise and adversarial attacks. Significant robustness emerges without specialized defences. Furthermore, SoftHebb tends to not merely be robust to attacks, but actually deflect them as specialized SOTA defences aim to do.

Here, we explored SoftHebb's applicability on several datasets. We measured its accuracy on MNIST, Fashion-MNIST, and CIFAR-10 in preliminary results, and we reported a functional convolutional SoftHebb network that improves accuracy on the significantly harder dataset of CIFAR-10. The convolutional implementation could become the foundation for deeper networks and complex problems. Ultimately, this could provide insights into the role of WTA microcircuits in larger networks in cortex with localized receptive fields (Pogodin et al., 2021), similar to area V1 of cortex (Hubel & Wiesel, 1962). All in all, the algorithm has several properties that are individually interesting and novel, and worth future extension. Combined, however, SoftHebb's properties shown in this work may already enable certain small-scale but previously-impossible applications. For example, fast, on-line, unsupervised learning of simple tasks by edge sensing devices, operating in noisy conditions, with a small battery and only local processing, requires those algorithmic properties that we demonstrated here.

## Reproducibility Statement

Some of our main contributions are theoretical and their proof is fully reproducible by following the rigorous derivations in the main text and in the appendices. Our experimental results are based on standard machine-learning techniques, while we also describe the hyperparameters and experimental protocols. Moreover, we provide Python code with specific instructions to reproduce the main experiments.

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

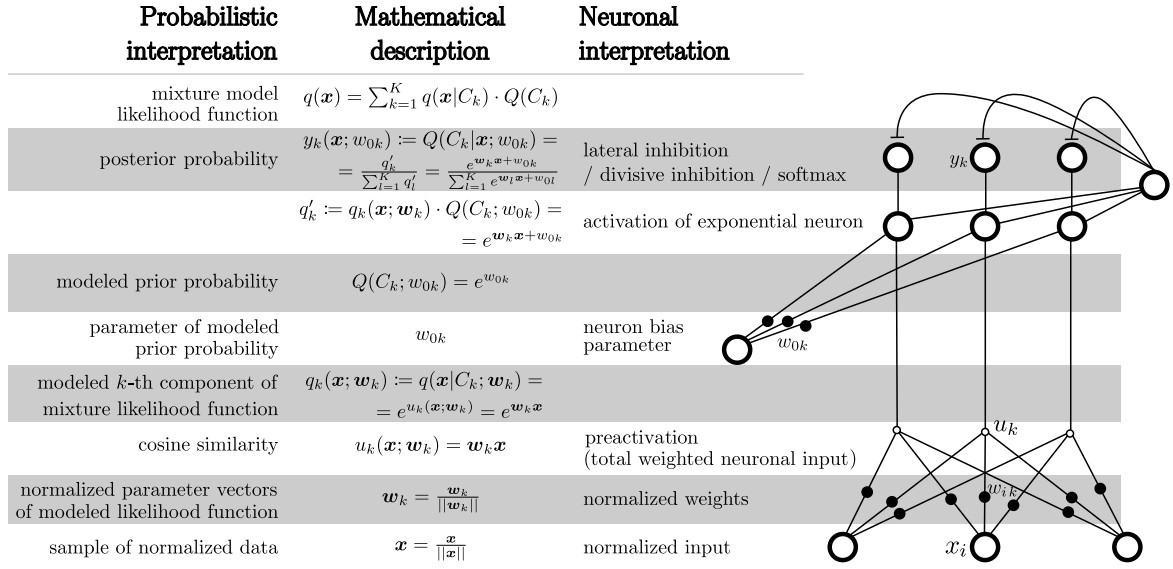

| Probabilistic interpretation | Mathematical description | Neuronal interpretation |
|---|---|---|
| mixture model likelihood function | $q(\boldsymbol{x}) = \sum_{k=1}^{K} q(\boldsymbol{x}|C_k) \cdot Q(C_k)$ | |
| posterior probability | $y_k(\boldsymbol{x}; w_{0k}) := Q(C_k|\boldsymbol{x}; w_{0k}) =$ $= \frac{q'_k}{\sum_{l=1}^{K} q'_l} = \frac{e^{\boldsymbol{w}_k \boldsymbol{x} + w_{0k}}}{\sum_{l=1}^{K} e^{\boldsymbol{w}_l \boldsymbol{x} + w_{0l}}}$ | lateral inhibition / divisive inhibition / softmax |
| | $q'_k := q_k(\boldsymbol{x}; \boldsymbol{w}_k) \cdot Q(C_k; w_{0k}) =$ $= e^{\boldsymbol{w}_k \boldsymbol{x} + w_{0k}}$ | activation of exponential neuron |
| modeled prior probability | $Q(C_k; w_{0k}) = e^{w_{0k}}$ | |
| parameter of modeled prior probability | $w_{0k}$ | neuron bias parameter |
| modeled $k$-th component of mixture likelihood function | $q_k(\boldsymbol{x}; \boldsymbol{w}_k) := q(\boldsymbol{x}|C_k; \boldsymbol{w}_k) =$ $= e^{u_k(\boldsymbol{x}; \boldsymbol{w}_k)} = e^{\boldsymbol{w}_k \boldsymbol{x}}$ | |
| cosine similarity | $u_k(\boldsymbol{x}; \boldsymbol{w}_k) = \boldsymbol{w}_k \boldsymbol{x}$ | preactivation (total weighted neuronal input) |
| normalized parameter vectors of modeled likelihood function | $\boldsymbol{w}_k = \frac{\boldsymbol{w}_k}{||\boldsymbol{w}_k||}$ | normalized weights |
| sample of normalized data | $\boldsymbol{x} = \frac{\boldsymbol{x}}{||\boldsymbol{x}||}$ | normalized input |

Figure 4: The soft WTA model (Moraitis et al., 2020) used in SoftHebb. The network graph is shown on the right. The input to the layer is shown at the bottom and the output is at the top. Each depicted computational element in the diagram is in a white or grey row that also includes the element's description on the left.

---

**Algorithm 1** SoftHebb learning

1: **for all** neurons $k \in \{1, 2, ..., K\}$ in the layer, **do**
2:      initialize random weights and biases

3: **for all** training examples $\boldsymbol{x}$ **do**
4:      **for all** neurons $k$ **do**
5:          Calculate preactivation $u_k = \boldsymbol{w}_k \boldsymbol{x}$
6:      **for all** neurons $k$ **do**
7:          Optional: calculate activation $q'_k = h(u_k + w_{0k})$          ▷ e.g. $h(x) = \exp(x)$
8:          Calculate posterior (i.e. normalized activation) $y_k$          ▷ e.g. Softmax
9:      **for all** neurons $k$ **do**
10:          **for all** synapses $i$ **do**
11:              calculate weight change $\Delta w_{ik}^{(SoftHebb)} := \eta \cdot y_k \cdot (x_i - u_k w_{ik})$
12:              update weight $w_{ik} \leftarrow w_{ik} + \Delta w_{ik}^{(SoftHebb)}$
13:          calculate bias change $\Delta w_{0k}^{SoftHebb} = \eta e^{-w_{0k}} (y_k - e^{w_{0k}})$
14:          update bias $w_{0k} \leftarrow w_{0k} + \Delta w_{0k}^{SoftHebb}$

---

***Proof of Theorem 2.3.*** The parameters of model $q$ are optimal $\boldsymbol{w} = {}_{opt}\boldsymbol{w}$ if they minimize the model's Kullback-Leibler divergence with the data distribution $p$. $D_{KL}(p(\boldsymbol{x})||q(\boldsymbol{x}; \boldsymbol{w}))$. Because $p_k := p(\boldsymbol{x}|C_k)$ is independent from $p_l$, and $q_k := q(\boldsymbol{x}|C_k; \boldsymbol{w}_k)$ is independent from $\boldsymbol{w}_l$ for every $l \neq k$, we can find the set of parameters that minimize the KL divergence of the mixtures, by minimizing the KL divergence of each component $k$: $\min D_{KL}(p_k||q_k)$, $\forall k$, and simultaneously setting

$$P(C_k) = Q(C_k; w_{0k}), \forall k. \qquad (12)$$

From Eq. 3 and this last condition, Eq. 5 of the Theorem is proven:

$$_{opt}w_{0k} = \ln P(C_k).$$

Further,

$$_{opt}\boldsymbol{w}_k := \arg\min_{\boldsymbol{w}_k} D_{KL}(p_k\|q_k)$$

$$= \arg\min_{\boldsymbol{w}_k} \int_{\boldsymbol{x}} p_k \ln \frac{p_k}{q_k} d\boldsymbol{x}$$

$$= \arg\min_{\boldsymbol{w}_k} \int_{\boldsymbol{x}} p_k \ln p_k - p_k \ln q_k d\boldsymbol{x}$$

$$= \arg\min_{\boldsymbol{w}_k} \int_{\boldsymbol{x}} -p_k \ln q_k d\boldsymbol{x} \tag{13}$$

$$= \arg\max_{\boldsymbol{w}_k} \int_{\boldsymbol{x}} p_k \ln q_k d\boldsymbol{x}$$

$$= \arg\max_{\boldsymbol{w}_k} \int_{\boldsymbol{x}} p_k \ln e^{u_k} d\boldsymbol{x} \tag{14}$$

$$= \arg\max_{\boldsymbol{w}_k} \int_{\boldsymbol{x}} p_k u_k d\boldsymbol{x}$$

$$= \arg\max_{\boldsymbol{w}_k} \mu_{p_k}(u_k)$$

$$= \arg\max_{\boldsymbol{w}_k} \mu_{p_k}(\cos(\boldsymbol{w}_k, \boldsymbol{x})). \tag{15}$$

where we used for Eq. 13 the fact that $\int_{\boldsymbol{x}} p_k \ln p_k d\boldsymbol{x}$ is a constant because it is determined by the environment's data and not by the model's parametrization on $\boldsymbol{w}$. Eq. 14 follows from the definition of $q_k$. The result in Eq. 15 is the mean value of the cosine similarity $u_k$.

Due to the symmetry of the cosine similarity, it follows that

$$_{opt}\boldsymbol{w}_k = \arg\max_{\boldsymbol{w}_k} \mu_{p_k}(\cos(\boldsymbol{w}_k, \boldsymbol{x})) = \arg\max_{\boldsymbol{w}_k} \cos(\boldsymbol{w}_k, \mu_{p_k}(\boldsymbol{x})) \tag{16}$$

$$= c \cdot \mu_{p_k}(\boldsymbol{x}), c \in \mathbb{R}. \tag{17}$$

Enforcement of the requirement for normalization of the vector leads to the unique solution $_{opt}\boldsymbol{w}_k^* = \frac{\mu_{p_k}(\boldsymbol{x})}{||\mu_{p_k}(\boldsymbol{x})||}$. $\qquad\square$

***Proof of Theorem 2.5***. We will find the equilibrium point of the SoftHebb plasticity rule, i.e. the weight $w_{ik}$ that implies $E[\Delta w_{ik}^{(SoftHebb)}] = 0$.

We will expand this expected value based on the plasticity rule itself, and on the probability distribution of the input $\boldsymbol{x}$.

$$E[\Delta w_{ik}^{(SoftHebb)}] = \eta \int_{\boldsymbol{x}} y_k(\boldsymbol{x}) \cdot (x_i - u_k(\boldsymbol{x})w_{ik})p(\boldsymbol{x})d\boldsymbol{x} \tag{18}$$

$$= \eta \int_{\boldsymbol{x}} y_k(\boldsymbol{x})(x_i - \boldsymbol{w}_k\boldsymbol{x}w_{ik}) \left(\sum_{l=1}^{K} p_l(\boldsymbol{x})P(C_l)\right) d\boldsymbol{x} \tag{19}$$

$$= \eta \left[\sum_{l=1}^{K} \int_{\boldsymbol{x}} x_i y_k(\boldsymbol{x})p_l(\boldsymbol{x})P(C_l)d\boldsymbol{x} - \sum_{l=1}^{K} \boldsymbol{w}_k w_{ik} \int_{\boldsymbol{x}} \boldsymbol{x}y_k(\boldsymbol{x})p_l(\boldsymbol{x})P(C_l)d\boldsymbol{x}\right]. \tag{20}$$

Based on this, we will now show that

$$\left[\boldsymbol{w}_k = {}_{opt}\boldsymbol{w}_k^* = \frac{\mu_{p_k}(\boldsymbol{x})}{||\mu_{p_k}(\boldsymbol{x})||} \text{ and } w_{0k} = {}_{opt}w_{0k}, \forall k\right] \tag{21}$$

$$\implies E[\Delta w_{ik}^{(SoftHebb)}] = 0 \quad \forall i, k. \tag{22}$$

Using the premise 21, we can take the following steps, where steps 3, 4, and 6 are the main ones, and steps 1, 2, and 5, as well as Theorem A.1 support those.

1. The cosine similarity function $u_k(\boldsymbol{x}) = \boldsymbol{x}\boldsymbol{w}_k$, as determined by the total weighted input to the neuron $k$, and appropriately normalized, defines a probability distribution centred symmetrically around the vector $\boldsymbol{w}_k$, i.e. $\mu_{u_k}(\boldsymbol{x}) = \boldsymbol{w}_k$ and $u_k(\mu_{u_k} - \boldsymbol{x}) = u_k(\mu_{u_k} + \boldsymbol{x})$.

   $\boldsymbol{w}_k$, as premised, is equal to the normalized $\mu_{p_k}(\boldsymbol{x})$, i.e. the mean of the distribution $p_k(\boldsymbol{x}) = p(\boldsymbol{x}|C_k)$, therefore: $\mu_{u_k}(\boldsymbol{x}) = \mu_{p_k}(\boldsymbol{x})$.

2. The soft-WTA version of the neuronal transformation, i.e. the softmax version of the model's inference is

$$y_k(\boldsymbol{x}) = \frac{\exp\left(u_k(\boldsymbol{x})\right)\exp(w_{0k})}{\sum_{l=1}^{K}\exp\left(u_l(\boldsymbol{x})\right)\exp(w_{0l})}, \tag{23}$$

But because of the premise 21 that the parameters of the model $u_k$ are set to their optimal value, it follows that $\exp(u_k(\boldsymbol{x})) = p_k(\boldsymbol{x})$ and $\exp(w_{0k}) = P(C_k)$, $\forall k$ (see also Theorem 2.6), therefore

$$y_k(\boldsymbol{x}) = \frac{p_k(\boldsymbol{x})P(C_k)}{\sum_{l=1}^{K} p_l(\boldsymbol{x})P(C_l)}. \tag{24}$$

3. Eq. 20 involves twice the function $y_k(\boldsymbol{x})p(\boldsymbol{x}) = \sum_{l=1}^{K} y_k(\boldsymbol{x})p_l(\boldsymbol{x})P(C_l)$. Using the above two points, we will now show that this is approximately equal to $y_k(\boldsymbol{x})p_k(\boldsymbol{x})P(C_k)$.

   (a) We assume that the "support" $\mathbb{O}_k^p$ of the component $p_k(\boldsymbol{x})$, i.e. the region where $p_k$ is not negligible, is not fully overlapping with that of other components $p_l$.

   In addition, $\mathbb{O}_k^p$ is narrow relative to the input space $\forall k$, because, first, the cosine similarity $u_k(\boldsymbol{x})$ diminishes fast from its maximum at $\arg\max u_k = \boldsymbol{w}_k$ in case of non-trivial dimensionality of the input space, and, second, $p_k(\boldsymbol{x}) = \exp\left(u_k(\boldsymbol{x})\right)$ applies a further exponential decrease.

   Therefore, the overlap $\mathbb{O}_k^p \cap \mathbb{O}_l^p$ is small, or none, $\forall l \neq k$.

   If $\mathbb{O}_k^y$ is the "support" of $y_k$, then this is even narrower than $\mathbb{O}_k^p$, due to the softmax. As a result, the overlap $\mathbb{O}_k^y \cap \mathbb{O}_l^p$ of $y_k$ and $p_l$ is even smaller than the overlap $\mathbb{O}_k^p \cap \mathbb{O}_l^p$ of $p_k$ and $p_l$, $\forall l \neq k$.

   (b) Because of the numerator in Eq. 24, the overlap $\mathbb{O}_k^y \cap \mathbb{O}_k^p$ of $y_k$ and $p_k$ is large.

   Based on these two points, the overlaps $\mathbb{O}_k^y \cap \mathbb{O}_l^p$ can be neglected for $l \neq k$, and it follows that

$$\sum_{l=1}^{K} y_k(\boldsymbol{x})p_l(\boldsymbol{x})P(C_l) \approx y_k(\boldsymbol{x})p_k(\boldsymbol{x})P(C_k). \tag{25}$$

Therefore, we can write Eq. 20 as

$$E[\Delta w_{ik}^{(SoftHebb)}]$$

$$\approx \eta \left[ \int_{\boldsymbol{x}} x_i y_k(\boldsymbol{x})p_k(\boldsymbol{x})P(C_k)d\boldsymbol{x} - \boldsymbol{w}_k w_{ik} \int_{\boldsymbol{x}} \boldsymbol{x} y_k(\boldsymbol{x})p_k(\boldsymbol{x})P(C_k)d\boldsymbol{x} \right]. \tag{26}$$

Next, we aim to show that the integrals $\int_{\boldsymbol{x}} x_i y_k(\boldsymbol{x})p_k(\boldsymbol{x})d\boldsymbol{x}$ and $\int_{\boldsymbol{x}} \boldsymbol{x} y_k(\boldsymbol{x})p_k(\boldsymbol{x})d\boldsymbol{x}$ involved in that Equation equal the mean value of $x_i$ and $\boldsymbol{x}$ respectively according to the distribution $p_k$.

To show this, we observe that the integrals are indeed mean values of $x_i$ and $\boldsymbol{x}$ according to a probability distribution, and specifically the distribution $y_k p_k$. We will first show that the distribution $p_k$ is symmetric around its mean $\mu_{p_k}(\boldsymbol{x})$. Then we will show that $y_k(\boldsymbol{x})$ is also symmetric around the same mean. Then we will use the fact that the product of two such symmetric distributions with common mean, such as $y_k p_k$, is a distribution with the same mean, a fact that we will prove in Theorem A.1.

4. Because the cosine similarity function $u_k(\boldsymbol{x})$ is symmetric around the mean value $\mu_{u_k}(\boldsymbol{x}) = \boldsymbol{w}_k$: $u_k(\mu_{u_k} - \boldsymbol{x}) = u_k(\mu_{u_k} + \boldsymbol{x})$, it follows that $p_k(\mu_{u_k} - \boldsymbol{x}) = \exp(u_k(\mu_{u_k} - \boldsymbol{x})) = \exp(u_k(\mu_{u_k} + \boldsymbol{x})) = p_k(\mu_{u_k} + \boldsymbol{x})$.

Therefore, $p_k(\boldsymbol{x}) = \exp\left(u_k(\boldsymbol{x})\right)$ does have the sought property of symmetry, around $\mu_{p_k}(\boldsymbol{x})$.

In point 1 of this list we have also shown that $\mu_{u_k}(\boldsymbol{x}) = \mu_{p_k}(\boldsymbol{x})$, thus $p_k$ is symmetric around its own mean $\mu_{p_k}(\boldsymbol{x})$.

5. The reason why the softmax output $y_k$ is symmetric around the same mean as $p_k$ consists in the following arguments:

   (a) The numerator $p_k$ of the $y_k$ softmax in Eq. 24 is symmetric around $\mu_{p_k}(\boldsymbol{x})$, as was shown in the preceding point.

   (b) The denominator of Eq. 24, i.e. $\sum_{l=1}^{K} \exp\left(p_l(\boldsymbol{x})\right) P(C_l)$ is also symmetric around $\mu_{p_k}(\boldsymbol{x})$ in $\mathbb{O}_k^p$, where $\mathbb{O}_k^p$ is the "support" of the $p_k$ distribution, i.e. the region where the numerator $p_k$ is not negligible.

   This is because:

      i. We assume that the data is distributed on the unit hypersphere of the input space according to $p(\boldsymbol{x})$ without a bias. Therefore the total contribution of components $\sum_{l \neq k} p_l(\boldsymbol{x}) P(C_l)$ to $p(\boldsymbol{x})$ in that neighbourhood $\mathbb{O}_k^p$ is approximately symmetric around $\mu_{p_k}(\boldsymbol{x})$.

      ii. $\sum_{l \neq k} p_l(\boldsymbol{x}) P(C_l)$ is not only approximately symmetric, but also its remaining asymmetry has a negligible contribution to $p(\boldsymbol{x})$ in $\mathbb{O}_k^p$, because $p(x)$ in $\mathbb{O}_k^p$ is mostly determined by $p_k(\boldsymbol{x})$.

   This is true, because as we showed in point 3a, $\mathbb{O}_k^p \cap \mathbb{O}_l^p$ is a narrow overlap $\forall l \neq k$.

      iii. $p_k$ is also symmetric, therefore, the total sum $\sum_{l=1}^{K} \exp\left(p_l(\boldsymbol{x})\right) P(C_l)$ is symmetric around $\mu_{p_k}(\boldsymbol{x})$.

   (c) The inverse fraction $\frac{1}{f}$ of a symmetric function $f$ is also symmetric around the same mean, therefore the inverse of the denominator $\frac{1}{p(\boldsymbol{x})} = \frac{1}{\sum_{l=1}^{K} \exp(p_l(\boldsymbol{x})) P(C_l)}$ is also symmetric around the mean value $\mu_{p_k}(\boldsymbol{x})$.

   (d) The normalized product of two distributions that are symmetric around the same mean is a probability distribution with the same mean. We prove this formally in Theorem A.1 and in its Proof at the end of the present Appendix A.

   Therefore $y_k = p_k \frac{1}{p(\boldsymbol{x})}$ is indeed symmetric around $\mu_{y_k}(\boldsymbol{x}) = \mu_{u_k}(\boldsymbol{x}) = \boldsymbol{w}_k = \mu_{p_k}(\boldsymbol{x})$.

In summary,

$$\mu_{p_k}(\boldsymbol{x}) = \mu_{y_k}(\boldsymbol{x}), \tag{27}$$

and, due to the symmetry,

$$p_k(\mu_{p_k} + \boldsymbol{x}) = p_k(\mu_{p_k} - \boldsymbol{x}), \tag{28}$$

$$y_k(\mu_{p_k} + \boldsymbol{x}) = y_k(\mu_{p_k} - \boldsymbol{x}). \tag{29}$$

6. Because the means of $y_k(\boldsymbol{x})$ and of $p_k(\boldsymbol{x})$ are both equal to $\mu_{p_k}(\boldsymbol{x})$, and because both distributions are symmetric around that mean, the probability distribution $y_k(\boldsymbol{x}) p_k(\boldsymbol{x}) P(C_k)/I_k$, where $I_k$ is the normalization factor, also has a mean $\mu_{y_k p_k}(\boldsymbol{x})$ equal to $\mu_{p_k}(\boldsymbol{x})$:

$$\int_{\boldsymbol{x}} \boldsymbol{x} y_k(\boldsymbol{x}) p_k(\boldsymbol{x}) P(C_k)/I_k \, d\boldsymbol{x} = \mu_{p_k}(\boldsymbol{x}). \tag{30}$$

We prove this formally in Theorem A.1 and in its Proof at the end of the present Appendix A.

Therefore, the **first component** of the sum in Eq. 26 is

$$\int_{\boldsymbol{x}} x_i y_k(\boldsymbol{x}) p_k(\boldsymbol{x}) P(C_k) d\boldsymbol{x} = I_k \cdot \mu_{p_k}(x_i) \tag{31}$$

and, similarly, the **second component** is

$$- \boldsymbol{w}_k w_{ik} \int_{\boldsymbol{x}} \boldsymbol{x} y_k(\boldsymbol{x}) p_k(\boldsymbol{x}) P(C_k) d\boldsymbol{x} = -I_k \boldsymbol{w}_k w_{ik} \cdot \mu_{p_k}(\boldsymbol{x}). \tag{32}$$

From the above conclusions about the two components of the sum in Eq. 26, it follows that

$$E[\Delta w_{ik}^{(SoftHebb)}] = I_k \cdot \mu_{p_k}(x_i) - I_k \boldsymbol{w}_k w_{ik} \cdot \mu_{p_k}(\boldsymbol{x}) \tag{33}$$

$$= I_k \cdot (\mu_{p_k}(x_i) - \boldsymbol{w}_k \mu_{p_k}(\boldsymbol{x}) \cdot w_{ik}) \tag{34}$$

$$= I_k \cdot (\mu_{p_k}(x_i) - ||\mu_{p_k}(\boldsymbol{x})|| \cdot w_{ik}) \tag{35}$$

$$= I_k \cdot \left( \mu_{p_k}(x_i) - ||\mu_{p_k}(\boldsymbol{x})|| \frac{\mu_{p_k}(x_i)}{||\mu_{p_k}(\boldsymbol{x})||} \right) \tag{36}$$

$$= 0. \tag{37}$$

Therefore, it is indeed true that $\left[ \boldsymbol{w}_k = {}_{opt}\boldsymbol{w}_k^* = \frac{\mu_{p_k}(\boldsymbol{x})}{||\mu_{p_k}(\boldsymbol{x})||} \forall k \right] \implies E[\Delta w_{ik}^{(SoftHebb)}] = 0 \, \forall i, k.$

Thus, the optimal weights of the model ${}_{opt}\boldsymbol{w}_k^* = \frac{\mu_{p_k}(\boldsymbol{x})}{||\mu_{p_k}(\boldsymbol{x})||} \forall k$ are equilibrium weights of the SoftHebb plasticity rule and network.

However, it is not yet clear that the weights that are normalized to a unit vector are those that the rule converges to, and that other norms of the vector are unstable. We will now give an intuition, and then prove that this is the case.

The multiplicative factor $u_k$ is common between our rule and Oja's rule (Oja, 1982). The effect of this factor is known to normalize the weight vector of each neuron to a length of one (Oja, 1982), as also shown in similar rules with this multiplicative factor (Krotov & Hopfield, 2019). We prove that this is the effect of the factor also in the SoftHebb rule, separately in Theorem A.2 and its Proof, provided at the end of the present Appendix A.

This proves Theorem 2.5, and satisfies the optimality condition derived in Theorem 2.3. $\qquad\square$

***Proof of Theorem 2.6.*** Similarly to the Proof of Theorem 2.5, we find the equilibrium parameter $w_{0k}$ of the SoftHebb plasticity rule.

$$E[\Delta w_{0k}] = \eta \int_{y_k} \left( y_k e^{-w_{0k}} - 1 \right) p(y_k) dy_k \tag{38}$$

$$= \eta \left( e^{-w_{0k}} \int_{y_k} p(y_k) dy_k - 1 \right) \tag{39}$$

$$= \eta \left( e^{-w_{0k}} E[y_k] - 1 \right) \tag{40}$$

$$= \eta \left[ e^{-w_{0k}} \mu_{p_k} \left( Q(C_k | \boldsymbol{x}) \right) - 1 \right] \tag{41}$$

$$= \eta \left[ e^{-w_{0k}} \mu_{p_k} \left( P(C_k | \boldsymbol{x}) \right) - 1 \right] \tag{42}$$

$$= \eta \left[ e^{-w_{0k}} P(C_k) - 1 \right] \tag{43}$$

Therefore,

$$E[\Delta w_{0k}] = 0 \implies$$
$$w_{0k}^{SoftHebb} = \ln P(C_k), \tag{44}$$

which proves Theorem 2.6 and shows the SoftHebb plasticity rule of the neuronal bias finds the optimal parameter of the Bayesian generative model as defined by Eq. 5 of Theorem 2.3. $\qquad\square$

**Theorem A.1.** *Given two probability density functions (PDF) $y(x)$ and $p(x)$ that are both centred symmetrically around the same mean value $\mu$, their product $y(x)p(x)$, normalized appropriately, is a PDF with the same mean, i.e.*

$$\begin{cases} \int_x xy(x)dx = \mu \\ \int_x xp(x)dx = \mu \\ p(\mu + x) = p(\mu - x) \\ y(\mu + x) = y(\mu - x) \end{cases} \implies \frac{1}{I} \int_x xy(x)p(x)dx = \mu. \tag{45}$$

*Proof of Theorem A.1.*

$$\int_{-\infty}^{+\infty} xy(x)p(x)dx = \int_{-\infty}^{\mu} xy(x)p(x)dx + \int_{\mu}^{+\infty} xy(x)p(x)dx \tag{46}$$

$$I_1 := \int_{-\infty}^{\mu} xy(x)p(x)dx \tag{47}$$

$$= \int_{+\infty}^{0} (\mu - u)y(\mu - u)p(\mu - u)d(\mu - u) \tag{48}$$

$$= -\int_{+\infty}^{0} (\mu - u)y(\mu - u)p(\mu - u)du \tag{49}$$

$$= -\int_{+\infty}^{0} \mu y(\mu - u)p(\mu - u)du + \int_{+\infty}^{0} uy(\mu - u)p(\mu - u)du \tag{50}$$

$$= -\int_{-\infty}^{0} \mu y(\mu + u)p(\mu + u)d(-u) - \int_{0}^{+\infty} uy(\mu - u)p(\mu - u)du \tag{51}$$

$$= \int_{-\infty}^{0} \mu y(\mu + u)p(\mu + u)du - \int_{0}^{+\infty} uy(\mu + u)p(\mu + u)du \tag{52}$$

$$= \mu \int_{-\infty}^{\mu} y(x)p(x)dx - \int_{0}^{+\infty} uy(\mu + u)p(\mu + u)du. \tag{53}$$

$$I_2 := \int_{\mu}^{+\infty} xy(x)p(x)dx \tag{54}$$

$$= \int_{0}^{+\infty} (\mu + u)y(\mu + u)p(\mu + u)d(\mu + u) \tag{55}$$

$$= \int_{0}^{+\infty} \mu y(\mu + u)p(\mu + u)du + \int_{0}^{+\infty} uy(\mu + u)p(\mu + u)du \tag{56}$$

$$= \mu \int_{\mu}^{+\infty} y(x)p(x)dx + \int_{0}^{+\infty} uy(\mu + u)p(\mu + u)du. \tag{57}$$

Therefore,

$$\int_{-\infty}^{+\infty} xy(x)p(x)dx = I_1 + I_2 \tag{58}$$

$$= \mu \int_{-\infty}^{\mu} y(x)p(x)dx - \int_{0}^{+\infty} uy(\mu + u)p(\mu + u)du \tag{59}$$

$$+\mu \int_{\mu}^{+\infty} y(x)p(x)dx + \int_{0}^{+\infty} uy(\mu + u)p(\mu + u)du = \mu \cdot I, \tag{60}$$

where $I = \int_{-\infty}^{+\infty} y(x)p(x)dx$.

$\square$

**Theorem A.2.** *The equilibrium weights of the SoftHebb synaptic plasticity rule of Eq. 8 are implicitly normalized by the rule to a vector of length 1.*

*Proof of Theorem A.2.* Using a technique similar to (Krotov & Hopfield, 2019), we write the Soft-Hebb plasticity rule as a differential equation

$$\tau \frac{dw_{ik}^{(SoftHebb)}}{dt} = \tau \Delta w_{ik}^{(SoftHebb)} = \tau \eta \cdot y_k \cdot (x_i - u_k w_{ik}). \tag{61}$$

The derivative of the norm of the weight vector is

$$\frac{d||\boldsymbol{w}_k||}{dt} = \frac{d(\boldsymbol{w}_k\boldsymbol{w}_k)}{dt} = 2\boldsymbol{w}_k\frac{d\boldsymbol{w}_k}{dt}. \tag{62}$$

Replacing $\frac{d\boldsymbol{w}_k}{dt}$ in this equation with the SoftHebb rule of Eq. 61, it is

$$\frac{d||\boldsymbol{w}_k^{SoftHebb)}||}{dt} = 2\frac{\eta}{\tau}\boldsymbol{w}_k \cdot y_k \cdot (\boldsymbol{x} - u_k\boldsymbol{w}_k) = 2\frac{\eta}{\tau}\boldsymbol{w}_k \cdot y_k \cdot (\boldsymbol{x} - \boldsymbol{w}_k\boldsymbol{x}\boldsymbol{w}_k)$$

$$= 2\frac{\eta}{\tau}u_ky_k \cdot (1 - ||\boldsymbol{w}_k||). \tag{63}$$

This differential equation shows that the derivative of the norm of the weight vector increases if $||\boldsymbol{w}_k|| < 1$ and decreases if $||\boldsymbol{w}_k|| > 1$, such that the weight vector tends a sphere of radius 1, which proves the Theorem. $\qquad\square$

APPENDIX B    DETAILS TO *Alternate activation functions* (SECTION 2.7)

Theorem 2.3, which concerns the synaptic plasticity rule in Eq. 8, was proven for the model of Definition 2.2, which uses a mixture of natural exponential component distributions, i.e. with base e (Eq. 4):

$$q_k := q(\boldsymbol{x}|C_k; \boldsymbol{w}_k) = e^{u_k}. \tag{64}$$

This implied an equivalence to a WTA neural network with natural exponential activation functions (Section 2.4). However, it is simple to show that these results can be extended to other model probability distributions, and thus other neuronal activations.

Firstly, in the simplest of the alternatives, the base of the exponential function can be chosen differently. In that case, the posterior probabilities that are produced by the model's Bayesian inference, i.e. the network outputs, $Q(C_k|\boldsymbol{x}; \boldsymbol{w}) = y_k(\boldsymbol{x}; \boldsymbol{w})$ are given by a softmax with a different base. If the base of the exponential is $b$, then

$$Q(C_k|\boldsymbol{x}; \boldsymbol{w}) = y_k = \frac{b^{u_k+w_{0k}}}{\sum_{l=1}^{K} b^{u_l+w_{0l}}}. \tag{65}$$

It is obvious in the Proof of Theorem 2.3 in Appendix A that the same proof also applies to the changed base, if we use the appropriate logarithm for describing KL divergence. Therefore, the optimal parameter vector does not change, and the SoftHebb plasticity rule also applies to the SoftHebb model with a different exponential base. This change of the base in the softmax bears similarities to the change of its exponent, in a technique that is called Temperature Scaling and has been proven useful in classification (Hinton et al., 2015).

Secondly, the more conventional type of Temperature Scaling, i.e. that which scales exponent, is also possible in our model, while maintaining a Bayesian probabilistic interpretation of the outputs, a neural interpretation of the model, and the optimality of the plasticity rule. In this case, the model becomes

$$Q(C_k|\boldsymbol{x}; \boldsymbol{w}) = y_k = \frac{e^{(u_k+w_{0k})/T}}{\sum_{l=1}^{K} e^{(u_l+w_{0l})/T}}. \tag{66}$$

The Proof of Theorem 2.3 in Appendix A also applies in this case, with a change in Eq. 14, but resulting in the same solution. Therefore, the SoftHebb synaptic plasticity rule is applicable in this case too. The solution for the neuronal biases, i.e. the parameters of the prior in the Theorem (Eq. 5), also remains the same, but with a factor of $T$: $_{opt}w_{0k} = T \ln P(C_k)$.

Finally, and most generally, the model can be generalized to use any non-negative and monotonically increasing function $h(x)$ for the component distributions, i.e. for the activation function of the neurons, assuming $h(x)$ is appropriately normalized to be interpretable as a probability density function. In this case the model becomes

$$Q(C_k|\boldsymbol{x}; \boldsymbol{w}) = y_k = \frac{h(u_k) \cdot w_{0k}}{\sum_{l=1}^{K} h(u_l) \cdot w_{0l}}. \tag{67}$$

Note that there is a change in the parametrization of the priors into a multiplicative bias $\boldsymbol{w}_0$, compared to the additive bias in the previous versions above. This change is necessary in this general case, because not all functions have the property $e^{a+b} = e^a \cdot e^b$ that we used in the exponential case. We can show that the optimal weight parameters remain the same as in the previous case of an exponential activation function, also for this more general case of activation $h$. It can be seen in the Proof of Theorem 2.3, that for a more general function $h(x)$ than the exponential, Eq. 14 would instead become:

$$_{opt}\boldsymbol{w}_k = \arg\min_{\boldsymbol{w}_k} D_{KL}(p_k||q_k) = \arg\max_{\boldsymbol{w}_k} \int_{\boldsymbol{x}} p_k \ln h(u_k) d\boldsymbol{x}$$

$$= \arg\max_{\boldsymbol{w}_k} \int_{\boldsymbol{x}} p_k \ln h(\cos(\boldsymbol{w}_k, \boldsymbol{x})) d\boldsymbol{x}$$

$$= \arg\max_{\boldsymbol{w}_k} \mu_{p_k} \left( \ln h(\cos(\boldsymbol{w}_k, \boldsymbol{x})) \right)$$

$$= \arg\max_{\boldsymbol{w}_k} \mu_{p_k} \left( g(\cos(\boldsymbol{w}_k, \boldsymbol{x})) \right), \tag{68}$$

where $g(x) = \ln h(x)$. We have assumed that $h$ is an increasing function, therefore $g$ is also increasing. The cosine similarity is symmetrically decreasing as a function of $\boldsymbol{x}$ around $\boldsymbol{w}_k$. Therefore, the function $g'(\boldsymbol{x}) = g(\cos(\boldsymbol{w}_k, \boldsymbol{x}))$ also decreases symmetrically around $\boldsymbol{w}_k$. Thus, the mean of that function $g'$ under the probability distribution $p_k$ is maximum when $\mu_{p_k} = \boldsymbol{w}_k$. As a result, Eq. 68 implies that in this more general model too, the optimal weight vector is $_{opt}\boldsymbol{w}_k = c \cdot \mu_{p_k}(\boldsymbol{x}), c \in \mathbb{R}$, and, consequently, it is also optimized by the same SoftHebb plasticity rule.

The implication of this is that the SoftHebb WTA neural network can use activation functions such as rectified linear units (ReLU), or other non-negative and increasing activations, such as rectified polynomials (Krotov & Hopfield, 2019) etc., and maintain its generative properties, its Bayesian computation, and the theoretical optimality of the plasticity rule. A more complex derivation of the optimal weight vector for alternative activation functions, which was specific to ReLU only, and did not also derive the associated long-term plasticity rule for our problem category (Definition 2.1), was provided by Moraitis et al. (2020).

## APPENDIX C  CROSS-ENTROPY AND TRUE CAUSES, AS OPPOSED TO LABELS

It is important to note that, in labelled datasets, the labels that have been assigned by a human supervisor may not correspond exactly to the true causes that generate the data, which SoftHebb infers. For example, consider MNIST. The 10 labels indicating the 10 decimal digits do not correspond exactly to the true cause of each example image. In reality, the cause $C$ of each MNIST example in the sense implied by causal inference is not the digit cause itself, but a combination of a single digit cause $D$, which is the MNIST label, with one of many handwriting styles $S$. That is, the probabilistic model is such that in the Eq. $P(\boldsymbol{x}) = \sum_k P(\boldsymbol{x}|C_k)P(C_k)$ of Definition 2.1, the cause $C$ of each sample is dual, i.e. there exists a digit $D_d$ ($d \in [0, 9]$) and a style $S_s$ such that

$$P(C_k) \coloneqq P(C = C_k) = P(D_d)P(S_s) \neq P(D_d). \tag{69}$$

$$\text{and } P(D_d) = \sum_k P(C_k)P(D_d|C_k). \tag{70}$$

This is important for our unsupervised model. To illustrate this point, a network with $K$ competing neurons trained on MNIST may learn not to specialize to $K$ digits $D$, but rather to $K$ handwriting styles $S$ of one digit $D_d$, or in general $K$ combinations of digits with styles – combinations, which are the true causes $C$ that generate the data. This leads in this case to a mismatch between the labels $D$, and the true causes $C$ of the data. Therefore, given the labels and not the causes, it is not obvious which number $K$ should be chosen for the number of neurons. Practically speaking, $K$ can be chosen using common heuristics from cluster analysis. It is also not obvious how to measure the loss of the WTA network during the learning process, since the ground truth for causes $C$ is missing. We will now provide the theoretical tools for achieving this loss-evaluation based on the labels. Even though SoftHebb is a generative model, it can be used for discrimination of the input classes $C_k$, using Bayes' theorem. More formally, the proof of Theorem 2.3 involved showing that SoftHebb minimizes the KL divergence of the model $q(\boldsymbol{x})$ from the data $p(\boldsymbol{x})$. Based on this it can be shown that the algorithm also minimizes its cross-entropy $H_Q^{causes} \coloneqq H(P(C), Q(C|\boldsymbol{x}))$ of the causes $Q(C_k|\boldsymbol{x})$ that it infers, from the true causes of the data $P(C_k)$: $\boldsymbol{w}^{SoftHebb} = arg\min_{\boldsymbol{w}} H_Q^{causes}$. An additional consequence is that by minimizing $H^{causes}$, SoftHebb also minimizes its label-based cross-entropy $H_Q^{labels} \coloneqq H(P(D_d), Q(D_d))$ between the true labels $P(D_d)$ and the implicitly inferred labels $Q(D_d)$:

$$Q(D_d) \coloneqq \sum_k Q(C_k)P(D_d|C_k) \tag{71}$$

$$\boldsymbol{w}^{SoftHebb} = arg\min_{\boldsymbol{w}} H_Q^{causes} = arg\min_{\boldsymbol{w}} H_Q^{labels}. \tag{72}$$

This is because, in Eqs. 70 and 71, the dependence of the labels on the true causes $P(D_d|C_k)$ is fixed by the data generation process. To obtain $Q(D_d|\boldsymbol{x})$ and measure the cross-entropy, the causal structure $P(D_d|C)$ is missing, but it can be represented by a supervised classifier $Q_2(D_d|Q(C|\boldsymbol{x}))$ of SoftHebb's outputs, trained using the labels $D_d$. Therefore, by (a) unsupervised training of SoftHebb, then (b) training a supervised classifier on top, and finally (c1) repeating the training of SoftHebb with the same initial weights and ordering of the training inputs, while (c2) measuring the trained classifier's loss, we can observe the cross-entropy loss $H^{labels}$ of SoftHebb while it is being

minimized, and infer that $H^{causes}$ is also minimized (Eq. 72). We call this the post-hoc cross-entropy method, and we have used it in our experiments (Section 3.2 and Fig. 1 C and D) to evaluate the learning process in a theoretically sound manner.

## APPENDIX D  DETAILS ON HEBBIAN EXPERIMENTS

In our validation experiments we found that softmax with a base of 1000 (see Section 2.7) performed best. The learning rate $\eta$ of Eq. 8 decreased linearly from 0.03 to 0 throughout training.

We found that an initial learning rate of 0.05 was best for the hard-WTA network. There are certain tunable options and hyperparameters for the training of SoftHebb as well as for its forward propagation to the next layer. Here we provide some guidelines for choosing the best options, and we report those that we used. We searched over the initial learning rate $\eta$ of Eq. 8 in the range of 0.1 and 0.001. The activation function (forward pass) was searched among RePU (rectified polynomial unit - a generalization of ReLU) **??**, softmax, sigmoid, tanh. For the 100 epochs experiment, we found that SoftHebb and its limit case of hard WTA are both stable with a learning rate that decays linearly to zero and starts between 0.08 and 0.02 while being optimum at an initial value of 0.045. As for the activation, we found that there are 3 accurate configurations; RePU with a power of 4.5, softmax with base of 200 and softmax with a smaller base of 1 if batch-normalization is added. For the 1 epoch run, the learning rate is best decaying exponentially, starting at 0.55 and ending at 0.0055. In that fast learning mode, only softmax activation with a high base of 200 gives a good accuracy.

For Fashion-MNIST the same options are best, except that an initial learning rate value of 0.065 performs better.

Supervised training of 100 epochs was done using Adam optimizer with mini-batch of 64. The learning-rate schedule starts at 0.001 and every 15 epochs is divided by 2.

In the single-epoch case, learning is online, i.e. mini-batch size is 1, for both the Hebbian and the supervised layer.

### D.1  CONVOLUTIONAL SOFTHEBB
We implemented a convolutional version of SoftHebb. In this case, the WTA soft competition is between convolutional kernels. This WTA computation and the corresponding SoftHebb update is iterated over each patch of the layer's input features, and this repeated operation over the patches is parallelized on GPUs. Effectively, this is similar to a fully connected SoftHebb that is trained on smaller patches of the original images.

## APPENDIX E  DETAILS ON ADVERSARIAL ATTACKS

We used "Foolbox", a Python library for adversarial attacks.

### E.1  TUNING PGD'S PARAMETERS

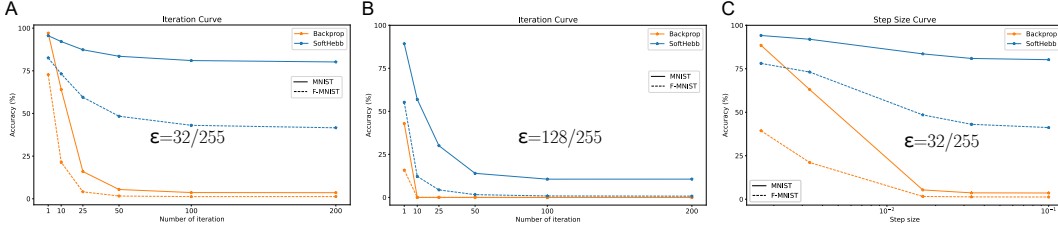

Figure 5: Noise and adversarial attack robustness of SoftHebb and of other unsupervised algorithms.

PGD has a few parameters that influence the effectiveness of the attack. Namely, $\epsilon$ which is a parameter determining the size of the perturbation, the number of iterations for the attack's gradient ascent, the step size per iteration, and a number of possible random restarts per attacked sample. Here we chose 5 random restarts. Then we found that 200 iterations are sufficient for both MNIST and

F-MNIST (Fig. 5 A and B). Then, using 200 iterations, and different $\epsilon$ values, we searched for a sufficiently good step size. We found that relative to $\epsilon$ a step size value of $\frac{0.01}{0.3}\epsilon \approx 0.33\epsilon$ (which is also the default value of the toolbox that we used) is a good value. An example curve for $\epsilon = 32/255$ is shown in Fig. 5 C for MNIST and F-MNIST.

## E.2    ADVERSARIAL ROBUSTNESS OF K-MEANS AND PCA

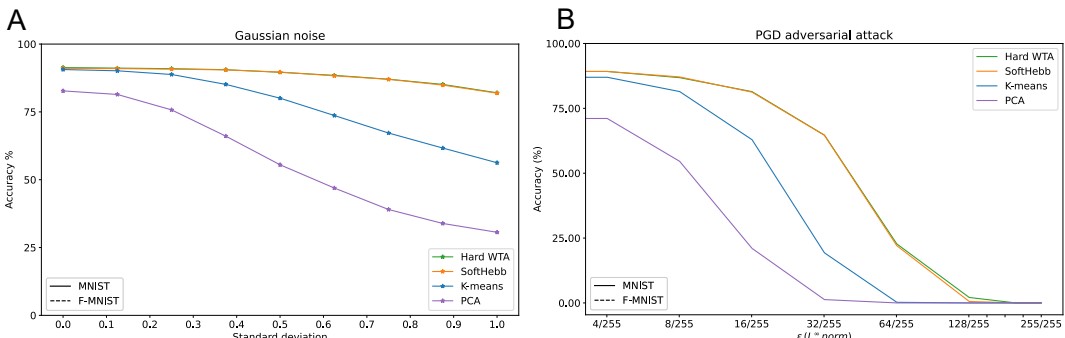

Figure 6: Noise and adversarial attack robustness of SoftHebb and of other unsupervised algorithms.

It is possible that the observed robustness of SoftHebb can be reproduced by other unsupervised learning rules. To test this possibility, we compared SoftHebb with PCA and k-means. We used 100 neurons, principal components, or centroids respectively. In PCA and k-means we then treated the learned coefficients as weight vectors of neurons and applied an activation function to then train a supervised classifier on top. First we attempted softmax as in SoftHebb. However the unperturbed test accuracy achieved at convergence was much lower. For example, on MNIST, k-means only reached an accuracy of $53.64\%$ and PCA $28.55\%$, whereas SoftHebb reached 91.06. Therefore, we performed the experiment again, but with ReLU activation for k-means and PCA, reaching $90.61\%$ and $82.74\%$ respectively. Then we tested for robustness, revealing that SoftHebb's learned features are in fact more robust than those of other unsupervised algorithms (Fig. 6). For completeness, we also include the hard WTA network, which is essentially a special case of SoftHebb with very high base in the softmax (Section 2.7), and is therefore learning equally robust features.

## E.3    EFFECT OF ACTIVATION FUNCTION ON ADVERSARIAL ROBUSTNESS

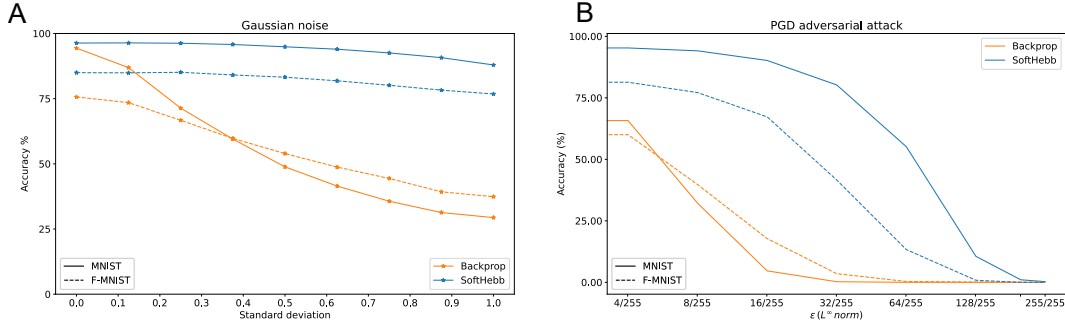

Figure 7: Noise and adversarial attack robustness of SoftHebb and of backpropagation-trained softmax-MLP on MNIST and Fashion-MNIST. Both SoftHebb and the MLP use a softmax activation at the hidden layer.

It is possible that the observed robustness of SoftHebb is due to the use of softmax as an activation function. To test this, we compared the SoftHebb network from Fig. 2 with a same-size backprop-trained 2-layer network, but here where the hidden layer's summed weighted input was passed through a softmax instead of ReLU before forwarding to the 2nd layer. First, we observed that at convergence, the backprop-trained network did not achieve SoftHebb's accuracy on either MNIST

(94.38%) or Fashion-MNIST (75.90%). Increasing the training time to 300 epochs did not help. Second, as can be seen in 7, backpropagation remains significantly less robust than SoftHebb to the input perturbations. This, together with the previous control experiments, suggests that, rather than its activation function, it is SoftHebb's learned representations that are responsible for the network's robustness.

