# OpenReview forum: "SoftHebb: Bayesian inference in unsupervised Hebbian soft winner-take-all networks"
_ICLR.cc/2022/Conference — ICLR 2022 Submitted_

### Official Review · Reviewer_4CvS · 2021-11-02

**Correctness:** 2
**Technical Novelty And Significance:** 2
**Empirical Novelty And Significance:** 2
**Recommendation:** 6
**Confidence:** 3

**Main Review:**

The paper mentions prior work in the same direction, such as (Nessler et al., 2009), but claims that a Bayesian theory for WTA learning has been missing. This claim is not correct.
The incomplete discussion of the state-of-the art makes it hard to identify the innovations of this paper.

Although the paper states that is aims at biological plausibility, its synaptic plasticity rule (8) is not local: the rule for adapting the weight from neuron i to neuron k requires knowledge of the current weights of all other synapses to the same postsynaptic neuron k (via the term u_k, see (4)).

A nice aspect of this paper is that it includes empirical studies of the noise robustness of the model. But it only compares the robustness of their model with that of MLPs, rather than with other unsupervised learning approaches.
..................
Some of these points have been improved in the update.



**Summary Of The Paper:**

The paper wants to provide an optimization theory for WTA networks that it claims to be missing. More precisely they consider soft WTA networks that are implemented as ANNs.They apply their model to MNIST and Fashion-MNIST. In addition some robustness to adversarial attacks is demonstrated.

**Summary Of The Review:**

The paper addresses an interesting topic. But it is not clear what its innovations are, and whether it can be implemented with a local learning rule.

---

> ### Author Response · Authors · 2021-11-18
> **Important objections by the reviewer, but only due to our earlier unclear text (part 1)**
>
> > The paper mentions prior work in the same direction, such as (Nessler et al., 2009), but claims that a Bayesian theory for WTA learning has been missing. This claim is not correct. The incomplete discussion of the state-of-the art makes it hard to identify the innovations of this paper.
>
> **Such a theory did _not_ exist for standard ANNs. Our writing may have caused this misunderstanding, and we have revised the manuscript to clarify this.**
>
> As we mentioned in the original manuscript but did not emphasize enough, thus far such theory exists only under quite limiting assumptions. Specifically, as you point out, we have cited Nessler et al. 2009; 2013, who did describe a very related and impactful theory but for a model that is very incompatible with ANNs and thus impractical. Neither the WTA layer nor the input are compatible with ANNs. First, both input and output neurons of the layer are spiking. Second, the neuronal activations in Nessler et al. are stochastic. Second, the inputs to Nessler et al.’s layer are discretized and require a population code. That is, each individual continuous variable such as an MNIST pixel, must be encoded as binary spikes of many neurons, with each neuron corresponding to a possible discrete value of pixel intensity. That work is foundational, has been very impactful in the SNN, computational neuroscience and neuromorphic literature, and provided inspiration for our paper. However, it does not show how to learn such a Bayesian framework without population codes, and without binary or stochastic neurons, i.e. it has limited practical value for ANNs. Essentially, it was not known if there is a plasticity rule that can learn to perform such Bayesian computation in a standard ANN, and, if so, which plasticity rule.
> This was an important missing piece, and our work provides it. This theory now enables the design and the understanding of Hebbian WTA ANNs through a Bayesian normative design.
>
> Moreover, our work extends the Bayesian WTA learning theory with further insights that were not provided by previous work even under the earlier impractical assumptions.
> Namely, we show that the probability distribution family of the underlying probabilistic model can be changed, but still maintain its Bayesian nature, and be implemented by a WTA ANN, and be learned by Hebbian plasticity. This is achieved through varying types of lateral inhibition, i.e. of normalization, and through varying activation functions.
>
> We thank the reviewer for bringing to our attention that this related SOTA was not sufficiently discussed, and that, as a result, our theoretic innovation was not made clear to the readers.
> In the previous version we had included the following quotes, but not prominently enough:
>
> From the previous Introduction section:
>
> “It remains unclear which specific plasticity rule and structure could optimize a non-spiking WTA for Bayesian inference, how to minimize a common loss function such as cross-entropy despite unsupervised learning, and how a cortical or artificial WTA could represent varying families of probability distributions.”
>
> In the paragraph after Definition 2.2:
>
> “Namely, (Nessler et al., 2009; 2013) considered data that was binary, and created by a population code, while the model was stochastic. These works provide the foundation of our derivation, but here we consider the more generic scenario where data are continuous-valued and input directly into the model, which is deterministic and, as we will show, more compatible with standard ANNs.”
>
> **We have now revised the manuscript to clarify this specific point, and have also included an extension of a dedicated paragraph upfront, in the introduction.** It is the second-to-last paragraph of the updated manuscript’s introduction. We moved a part from section 2.8 to Appendix C to make space for this and other planned additions.
>
> Further, our theoretical insights led us to reveal novel strengths of Hebbian WTA networks also in the experiments. This novelty of our experimental aspects is, we understand, clear to the reviewer.
>
> **We believe that through our clarifications, now also our theoretical innovation and its importance are pinpointed in a clear manner for the reviewer and the ICLR audience.**
>
> _continued in next comment..._

---

> > ### Author Response · Authors · 2021-11-18
> > **Important objections by the reviewer, but only due to our earlier unclear text (part 2)**
> >
> > _...continued from previous comment_
> > > Although the paper states that is aims at biological plausibility, its synaptic plasticity rule (8) is not local: the rule for adapting the weight from neuron i to neuron k requires knowledge of the current weights of all other synapses to the same postsynaptic neuron k (via the term u_k, see (4)).
> >
> > **The rule is in fact completely local. In the original submitted version of the manuscript, we did not provide clarifications on this point, which now we provide here and also in the revised manuscript.**
> >
> > The plasticity rule (Eq. 8 of the manuscript) that dictates the updates of weight from neuron $i$ to neuron $k$ is
> > $$
> > \Delta w^{(SoftHebb)}\_{ik} := \eta \cdot y_k \cdot (x_i − u_kw_{ik})
> > $$
> > As can be seen, all involved variables are local to the synapse, i.e. only indices $i$ and $k$ are relevant. No signals from distant layers, from non-perisynaptic neurons, or from other synapses are involved. **We have added this clarification to the revised manuscript**, immediately following Eq. 8.
> >
> > Knowing the term $u_k$ does not imply knowing the other weights $w_{jk}$. In the probabilistic (i.e. not the neuronal) view of the model, the definition of $u_k$ is given in Definition 2.2 of the manuscript. However, a full description in neuronal terms was missing before, and this may have caused the misunderstanding. **We now do clarify this in the revised manuscript**’s section 2.4, as well as here:
> >
> > $u_k$ is the total normalized weighted sum of the inputs to the neuron. For its update, there is no need for the synapse to know $x_j$ or $w_j$, i.e. variables pertaining to other synapses. The neuron-level summary $u_k$ is enough, and this variable is present at the immediately postsynaptic neuron of the synapse $i \rightarrow k$. In a biological interpretation, $u_k$ would be analogous to the total input current flowing into the soma of the synapse's local postsynaptic neuron, or to the membrane potential, so $u_k$ is indeed local.
> >
> > > The paper addresses an interesting topic. But it is not clear what its innovations are, and whether it can be implemented with a local learning rule.
> >
> > **That is the reviewer’s summary, and the two objections were important but misled by our earlier text. We believe that our revisions and clarifications here have addressed both objections.**

---

> > > ### Author Response · Authors · 2021-11-18
> > > **Response to Reviewer 4CvS (part 3)**
> > >
> > > The reviewer made another point besides their main points in their summary.
> > >
> > > > compares the robustness of their model with that of MLPs, rather than with other unsupervised learning approaches
> > >
> > > **The robustness comparison to backpropagation is quite novel and provides valuable new insights to multiple fields, opening indeed new research avenues.**
> > >
> > > Through these robustness comparisons, our work has revealed an important potential benefit of Hebbian WTA networks that was previously unknown. This opens a spectrum of new research questions, including indeed whether other unsupervised learning approaches also have such strengths. However, we suggest that the present first result is quite valuable in itself and that other researchers would benefit strongly from its publication. Would the communities of Hebbian-learning, adversarial-robustness, and cognitive-neuroscience benefit if the present results were kept unpublished? The robustness result in its present form is arguably quite intriguing, may lead to new out-of-the-box approaches to adversarial robustness, and it is not even the only valuable aspect of our experimental part, let alone our work's theoretical part.
> > >
> > > _Update:_ **We have now performed the requested robustness comparisons to other unsupervised algorithms**, and provide those in the revised manuscript. We have made a separate comment on this (https://openreview.net/forum?id=IJ-88dRfkdz&noteId=9PrbjPM1dMj) but are pointing it out here as well for clarity in the discussion.
> > >
> > > In summary, we believe that we have addressed all of the Reviewer's concerns.
> > >
> > > _edit:_ Our concluding remarks of the discussion with Reviewer 4CvS are in this separate comment: https://openreview.net/forum?id=IJ-88dRfkdz&noteId=1A-2aBAJ4eW

---

> ### Author Response · Authors · 2021-11-23
> **New experiments: robustness comparison with other unsupervised approaches**
>
> > A nice aspect of this paper is that it includes empirical studies of the noise robustness of the model. But it only compares the robustness of their model with that of MLPs, rather than with other unsupervised learning approaches.
>
> **We have now performed the comparison requested by the reviewer.**
> We have included an additional Appendix E with new experiments on adversarial robustness. Section E.2 compares the adversarial robustness of SoftHebb with that of k-means and PCA. SoftHebb is more robust than both alternatives.
>
> **We believe that we have now addressed all of the reviewer's concerns** through clarifications, new data, and revisions of the manuscript.
>
> _edit:_ Our concluding remarks of the discussion with Reviewer 4CvS are in this separate comment: https://openreview.net/forum?id=IJ-88dRfkdz&noteId=1A-2aBAJ4eW

---

> ### Author Response · Authors · 2021-11-30
> **All reviewer's points addressed**
>
> > .................. Some of these points have been improved in the update.
>
> We thank the reviewer for updating their score to recommend acceptance and editing their initial comment to include this acknowledgment.
>
> However, as we have described in the other comments, we have in fact **addressed _all_ the points that the reviewer raised initially**, not some of them. In fact one of the principal initial objections by the reviewer was factually wrong - a possible misunderstanding - claiming that the plasticity rule was not local. We are thankful that the reviewer did consider our comments and updates to clarify the understanding of our work. We also added all requested experiments to compare with other unsupervised learning methods.
>
> This discussion has left no remaining perceived weaknesses, and therefore we believe that the paper is ready for publication.

---

### Official Review · Reviewer_vMDj · 2021-11-02

**Correctness:** 4
**Technical Novelty And Significance:** 3
**Empirical Novelty And Significance:** Not applicable
**Recommendation:** 6
**Confidence:** 4

**Main Review:**

Strengths:
- The formalization of (a type of) competitive Hebbian learning from an optimization perspective is interesting and useful.
- The proposed method does seem to perform well.
- The experimental investigation on the effects of noise and adversarial attacks in the proposed method vs standard gradient-based methods is very nice.

Weaknesses:
- Evaluation is a bit weak (more below).
- Evaluation is performed only on two simple datasets, MNIST and Fashion-MNIST.
- The models evaluated only have 1 or 2 layers, which are extremely simplistic and limited. It is also not clear whether and how well the method would scale when a deep architecture is used.


**Summary Of The Paper:**

The authors pursue a formalization of WTA networks as optimization. In the paper, they show that WTA networks that compete via soft-max (approximating possibly plausible biological lateral inhibition) and are updated by Hebbian learning are implicitly minimizing the cross-entropy between inputs and layer activation.


**Summary Of The Review:**

The paper has both valid strengths and weaknesses. The paper could be a clear 'accept' for me if the authors would at least present results with more standard MNIST architectures, like conv+pool layers and/or 3-4+ layers. Additionally, it would be nice to see results of the method at least on CIFAR-10.

---

> ### Author Response · Authors · 2021-11-23
> **New experiments: CIFAR-10 and convolutions**
>
> > The paper could be a clear 'accept' for me if the authors would at least present results with more standard MNIST architectures, like conv+pool layers and/or 3-4+ layers. Additionally, it would be nice to see results of the method at least on CIFAR-10.
>
> **We have now indeed performed additional experiments, testing the algorithm on larger problems and implementing more complex architectures.**
>
> We now report the network’s accuracy on **CIFAR-10** (49.78%).
>
> In addition, **we have implemented a convolutional version** of SoftHebb, and our tests show that it is able to learn. It achieves in fact a much improved accuracy of 98.63% on MNIST and 60.30% on CIFAR-10.
>
> Moreover, we have revised the manuscript to include **a new section** titled “Extensibility of SoftHebb: F-MNIST, CIFAR-10, conv-SoftHebb” (Section 3.4).
>
> We believe that, together with our earlier comments and changes to the manuscript, **we have now fully addressed all of the reviewer’s concerns.**

---

> ### Author Response · Authors · 2021-11-29
> **Concluding summary of the discussion with Reviewer vMDj**
>
> > The paper could be a clear 'accept' for me if the authors would at least present results with more standard MNIST architectures, like conv+pool layers and/or 3-4+ layers. Additionally, it would be nice to see results of the method at least on CIFAR-10.
>
> We did follow the Reviewer's suggestion and provided results on CIFAR-10, and results from a convolutional implementation, along with further experiments, and with added discussion on extensibility.
>
> > The paper could be a clear 'accept' for me
>
> **Based on the Reviewer's earlier comment, the revised paper is now a clear 'accept'.** It is unfortunate that the Reviewer did not update their comments to reflect this. The Reviewer did not react in any way to our responses, but we do appreciate the initial feedback, which we fully used and addressed.

---

### Official Review · Reviewer_F6Fz · 2021-11-04

**Correctness:** 3
**Technical Novelty And Significance:** 2
**Empirical Novelty And Significance:** 2
**Recommendation:** 6
**Confidence:** 4

**Main Review:**

# UPDATED SCORE
* Score was updated to reflect the changes made by the authors during the rebuttal phase: 5-> 6

# Strengths
* Biologically-inspired Bayesian generative neural network that combines Hebbian learning with winner-takes-all connectivity
* Apparent improvement in robustness both to white-box adversarial attacks (PGD) and black-box attacks (Gaussian Noise)

# Weaknesses
* Model is evaluated only on MNIST and Fashion-MNIST datasets and is not clear whether it can be expanded to more challenging tasks
* Details on the adversarial attacks are missing
* Improvements other than those related to robustness are marginal at best and some controls are missing
* Detailed explanation on the model implementation is missing


# Detailed explanation of criticism
## MNIST and Fashion-MNIST only
The authors implement a SoftHebb model for the MNIST and Fashion-MNIST datasets. While these are interesting toy datasets, they are very simple to solve and solutions to them not always generalize to more challenging problems closer to real-world applications. In that sense the authors should do a better job at pointing a direction for future model development. For example, the authors claim that the number of true causes for the data, which in their single-layer implementation translates to number of neurons, can be chosen using common heuristics from cluster analysis. Can the authors indicate what this number would be for the CIFAR-10, TinyImageNet or even ImageNet datasets? How can this framework be adapted for a multilayer network? And can it be used in conjunction with convolutional layers which are a key component of neural networks for visual tasks? If not, wouldn’t it make more sense to demonstrate the use of this framework in a non-visual problem?

## Details on adversarial attacks are missing
Very interestingly, the SoftHebb model is very robust to both white-box and black-box attacks. While the robustness to Gaussian noise is undisputed (though it would be interesting to evaluate robustness to other types of random noise), the same cannot be said about the PGD adversarial attacks. When claiming improvements in robustness to gradient-based attacks, the burden of proving that the attacks have been properly performed is on the proponents of the defense. Unfortunately, in the current form of the paper, it is impossible to evaluate if the attack has been properly implemented. PGD has multiple hyperparameters, such as the number of the attack steps and the attack step size, that need to be optimized for different models separately. The authors make no reference to the choice of these hyperparameters and whether they tested different combinations to ensure that the attack was optimal. A bare minimum would be to show attack iteration curves for different attack step sizes for an intermediate perturbation strength. Also, it has been shown that the activation function greatly affects adversarial robustness (Xie et al 2021), does the standard MLP use the same activation function of the SoftHebb? It is possible that the improvements in robustness in SoftHebb are due to more trivial causes and do not depend necessarily on the proposed framework.

## Marginal improvements
The performance gains of the SoftHebb over the hard WTA model are marginal at best as well as the gains after 1 epoch over the standard backprop model. In the paper, I think a few visualizations are missing. For example, the authors should include the backprop model on Figure 1A and the hard WTA model on Figure 1B,C,D and Figure 2. This way, it gives the impression that the authors are cherry picking the baselines to compare based on the benchmark. While the authors make it clear in the main text that the backprop model outperforms their model in 100 epochs, comparisons between the SoftHebb and the hard WTA model for the remaining benchmarks are missing. How much better is the SoftHebb model over the hard WTA in terms of robustness?

## Improving model explanation
Theory could be better introduced/contextualized for example by illustrating how the different terms relate to the components of the ANNs. Also, the model description should be expanded either in the main text or supplementary. Diagram of the connectivity, formula for the neurons activation function as implemented in the code, and an algorithmic description of the model training would greatly improve understanding. Also, details on computational times are missing. While the authors focus their analyses on the number of epochs and training examples, it is not clear whether the training times and inference times of the different models is the same or deviate considerably, as well as the memory requirements to train/run each model.

## Minor points
* The authors point the dependence on feedback by standard ANNs as a criticism. However, there is a high degree of feedback in the brain. I suggest that the authors soften this point.


**Summary Of The Paper:**

In this paper, the authors develop a theoretical framework to incorporate winner-takes-all (WTA) connectivity with Hebbian-like plasticity which translates into a Bayesian generative model that can be used with generic artificial neural network (ANN) elements. This implementation, which they call SoftHebb, marginally outperforms a competing WTA alternative in digit classification on the MNIST. Furthermore, it also trains marginally faster than a standard backprop multilayer perceptron (MLP), though its performance saturates at a lower accuracy. Finally, the model appears to be substantially more robust than the standard MLP for both black-box and white-box attacks.

**Summary Of The Review:**

This study is definitely an interesting take on incorporating biologically-plausible components in an ANN design. However, the proposed model contains several limitations in terms of its expansibility to more challenging problems and some controls are missing to ensure the validity of the main claims, particularly those related to the robustness of the model. For these reasons, I cannot suggest accepting the paper in the current, but am willing to update the score if the authors address my concerns.

---

> ### Author Response · Authors · 2021-11-23
> **Response to Reviewer F6Fz**
>
> > Model is evaluated only on MNIST and Fashion-MNIST datasets and is not clear whether it can be expanded to more challenging tasks
>
> We now report new **results on CIFAR-10** and we describe **a convolutional implementation.**
>
> We have included **a new section 3.4 on extensibility** of the model
>
> > Marginal improvements
>
> We have **re-written sections** of the manuscript, clarifying that the model has **other key advantages compared to backpropagation than its accuracy**. In addition, the fact that it outperforms backpropagation even marginally is by itself astonishing. Please also read our other, more detailed comment on this topic: https://openreview.net/forum?id=IJ-88dRfkdz&noteId=wkRO7sZpBQT
>
> Moreover, we have now updated the plots, as the reviewer requested, to include comparisons between all models in each figure.
>
> > comparisons between the SoftHebb and the hard WTA model
>
> We have now added emphasis to the related part of the theoretical section 2.7. It now further emphasizes that our framework allows the hard WTA to be understood as a limit case of the same Bayesian framework, i.e. as a special case of SoftHebb. As a result, it is expected to have some of the benefits of a softer WTA, but also some differences.
>
> Further, we now do show the experimental comparison of  the two models in every subplot. To accompany this, we have re-written the experimental section, emphasizing those points from the experimental results that are key. A summary is that backpropagation is expectedly better than the unsupervised models, but, remarkably, not in the first epoch. A SoftHebb model is faster and more accurate than both a hard WTA and backpropagation, in the case where the data is only seen once by both layers. The Hebbian models are both equally robust and much more robust than backpropagation.
>
> Among our new re-focused experiments, we now perform an experiment that exploits the update-unlocked operation of SoftHebb. In simultaneous, i.e. not greedy, training of the two layers, the softhebb layer can receive a training input while the previous input has not yet updated the next layer. While these networks are commonly trained greedily, here we show that this operation that resolves the update-locking problem still performs better than backpropagation in the first epoch.
>
>  within our our clarifications and experiments showing that while SoftHabout the relation of hard WTA, several comparisons and analogies between the two cases both at
> > Details on adversarial attacks are missing
>
> We have now included an **Appendix E providing all the details and additional control experiments** requested by the reviewer, as well as robustness comparisons to other unsupervised models.
>
> > Improving model explanation
>
> Thank you for this suggestion, which we have followed and we believe has indeed significantly increased the clarity of the manuscript. At the top of the Appendix, we have **added a detailed diagram** with description of each element **in both its Bayesian probabilistic interpretation and in neural terms**.
>
> Following this, we have also included a **pseudo-code description of the algorithm** in the Appendix.
>
> > it is not clear whether the training times and inference times of the different models is the same or deviate considerably, as well as the memory requirements to train/run each model.
> We have implemented a custom library in PyTorch for SoftHebb that achieves good utilization of the GPU and therefore the model runs very fast. We have not performed systematic benchmarking of the model's computation compared to backpropagation, but it is comparable in terms of training time per epoch. Further, as we have hinted over this discussion and in the manuscript, SoftHebb's ultimate acceleration would be in custom hardware that exploits the principles of local signals, lack of symmetric forward-backward weights (simplifying the neuromorphic circuit), update-unlocked operation (no waiting for deeper layers), and lack of gradient computation, storage, or propagation. The result is likely to be much faster than backpropagation. However, this is futuristic at this stage.
>
> > Minor points
> The authors point the dependence on feedback by standard ANNs as a criticism. However, there is a high degree of feedback in the brain. I suggest that the authors soften this point.
>
> This is a fair point. We have followed the reviewer's suggestion and indeed softened our statements regarding feedback. However, we should point out that the relaxation of feedback-dependence remains an important advantage of approaches such as ours compared to backpropagation. We have added new clarifications to the introduction and the discussion describing the related issues of backpropagation that our approach solves, in addition to the performance advantages that we demonstrate. Those issues are also discussed in the comment we linked above.
>
> **We believe that we have now addressed fully all of the reviewer's comments**.

---

> > ### Comment · Reviewer_F6Fz · 2021-11-25
> > **Improved manuscript and updated score**
> >
> > I thank the authors for the extensive improvements they did to the manuscript which address most of my concerns. However, I still find the reported gains to be very weak, if any at all, particularly when considering the hard-WTA comparison. For this reason, I have now updated my score to a 6.

---

> > > ### Author Response · Authors · 2021-11-30
> > > **Advantages that the reviewer may have not considered**
> > >
> > > > address most of my concerns
> > >
> > > We thank the reviewer for the opportunity and the directions that we followed to improve the manuscript.
> > >
> > > > the reported gains to be very weak
> > >
> > > This added note by the reviewer appears to be focusing on certain advantages but missing the more important aspects of our work:
> > > 1. **The work is principally theoretical**. We provide a non-trivial and important missing piece of the literature.
> > > 2. SoftHebb **resolves several heavily researched inefficiencies and bio-implausibilities of backpropagation**: non-locality of the learning rule, update-locking, weight-transport, dependence on heavy and detailed feedback signals (see separate comment: https://openreview.net/forum?id=IJ-88dRfkdz&noteId=wkRO7sZpBQT).
> > > 3. On top of that, we do demonstrate accuracy advantages in certain conditions compared to backpropagation. Biological plausibility has been associated with disadvantages in terms of accuracy even in only partially plausible cases (see for example Feedback Alignment and other approaches cited in our introduction), let alone the high degree of biological plausibility that we achieve here. Showing an **advantage in accuracy is remarkable for the biologically-plausible learning literature, even if this advantage is small**.
> > > 4. The **gains in robustness are very significant**, and are a **radically different, emergent approach to robustness**.
> > >
> > > > particularly when considering the hard-WTA comparison
> > >
> > > Our latest revised manuscript makes clearer that **here, hard-WTA also uses the optimal plasticity rule that emerged from our SoftHebb theory.**
> > > In addition, our theoretical framework shows that a hard-WTA that uses our plasticity rule is in fact **a special case of SoftHebb**, where softmax uses a large base (see section 2.7 "Alternate activation functions and relation to hard WTA").
> > >
> > > Therefore, based on our optimality proofs, the generalized soft version was expected to perform similarly but better, which is what our experiments confirmed. The soft version is more accurate even though a supervised layer is added to the network and thus compensates for hard-WTA's disadvantage. If the Hebbian layer is studied individually, the soft version is in fact **a significantly faster unsupervised learner (Fig. 1D).**
> > >
> > > Therefore, our work integrates soft and hard WTA in a single theoretical framework, shows that the derived plasticity rule indeed works well for the hard version too, but also confirms the soft version's advantages that emerge from its theoretical optimality.
> > >
> > > **All in all, we believe that our work's results that the reviewer evaluated to already recommend acceptance are not insignificant, and this is supported by our data. More importantly, there are other, more significant practical advantages that the reviewer may have not considered. Lastly, the work has important theoretical value that the reviewer may also have neglected.** We hope that this comment further updates the reviewer's understanding of our work's value.

---

### Official Review · Reviewer_ereX · 2021-11-07

**Correctness:** 3
**Technical Novelty And Significance:** 2
**Empirical Novelty And Significance:** 2
**Recommendation:** 6
**Confidence:** 4

**Main Review:**


* The equivalency between mixture model and WTA has been proposed in a previous work (Moraitis et al. 2020) so the novelty on that end is quite limited.

* There are several other Hebbian and non-Hebbian learning rules that have been used in other works. Will any of them be adoptable in this framework?

* The comparisons are only made with 1 or 2-layer MLP and the accuracy differences are not significant. It is not clear what the advantage of softHebb is. The performance seems to be poor on the F-MNIST dataset. Discussion and experiments on learning with larger/more complex datasets need to be addressed.

* The adversarial robustness results seem promising compared to MLP, but a comparison is need with other adversarial learning/robustness techniques to assess the utility.


**Summary Of The Paper:**

The work presents an interesting equivalency between a generative probabilistic mixture model and a winner-take-all Hebbian local learning rule-based learning. The approach seems promising, especially the adversarial robustness results compared to a multi-layer perceptron.

**UPDATED SCORE**
I would like to thank the authors for their revisions and modification based on my feedback. I am changing my score from 5 --> 6 to reflect that. I would also like to mention that there have been works based on Hebbian/Non-Hebbian Learning that outperform backdrop-based  approaches in challenging scenarios such as continual supervised/reinforcement learning. There are other strategies for backprop networks such as implicit formulation (treating NN as a dynamical system) that improves adversarial robustness without adversarial training. Moreover, from a practical usability standpoint, even though SoftHebb is faster, the accuracy still trails the backprop-based networks.



**Summary Of The Review:**

The approach is promising, but is limited in terms of novelty, experiments and the comparison with existing approaches that makes it difficult to assess the true potential of this approach.

---

> ### Author Response · Authors · 2021-11-20
> **Response to Reviewer ereX (part 1)**
>
> > mixture model and WTA has been proposed in a previous work (Moraitis et al. 2020) so the novelty on that end is quite limited
>
> **That is true, but that is not our claimed novelty.**
>
> The reviewer is correct about the previously-shown equivalence between mixture model and WTA in Moraitis et al. 2020, and we have cited that and other related works. However, our result is about learning. It was not previously known how to learn that Bayesian mixture model in an ANN, or that it was possible with a Hebbian-like local rule, or which plasticity rule that should be precisely. That is the main theoretical contribution of our work. It is not at all a trivial extension of the prior literature, as evidenced by the proofs in our Appendix. In addition, it is an important result, and it is directly useable, as we show in the experiments.
>
> > There are several other Hebbian and non-Hebbian learning rules that have been used in other works. Will any of them be adoptable in this framework?
>
> **Not any learning rule can be used. Proving which rule can be used is our key theoretical novelty.** For example, earlier work found a plasticity rule that achieves something similar in a model of stochastic neurons and other constraints that limit its practical usability. That plasticity rule in our more conventionally testable model, would not learn the Bayes-optimal weights. However, other rules may in fact result in the same model, but with other trade-offs. For instance, a non-Hebbian rule, e.g. a supervised one such as the delta rule can be used to minimize the cross entropy of the layer’s outputs and the true causes of the data, if these causes are provided as labels. Our Hebbian-like rule provably achieves that same loss minimization without using supervisory labels in the training set.
>
> > It is not clear what the advantage of softHebb is
>
> > larger/more complex datasets
>
> > poor on the F-MNIST
>
> There are **multiple advantages in SoftHebb**, and significant, non-trivial value in our experiments. We would like to refer the reviewer to our comment that addresses these concerns: https://openreview.net/forum?id=IJ-88dRfkdz&noteId=wkRO7sZpBQT There, we clarify the value of such small-scale results from biologically-plausible learning, and describe our work’s advances compared to the state of the art. In summary:
> - SoftHebb **overcomes several important inefficiencies of backpropagation** SOTA, a key goal of neuromorphic computing.
> - Our work is the **first to present even small accuracy advantages** of biological learning compared to backpropagation. Even if only in small-scale conditions, this is remarkable.
> - Such small-scale problems solved as efficiently as we do through SoftHebb, **can in fact find real-world use** in intelligent edge-devices that locally solve small learning tasks robustly
> - Directly applying Hebbian learning in multilayer networks to approach complex problems would not be productive without a **first understanding of the theory and emergent properties of the single layer**. This previously missing prerequisite is significant and non-trivial, and therefore demanded a dedicated study before possible larger networks. _update: We now apply SoftHebb to more complex data & even to a convolutional architecture (see below)._
>
> Moreover, **we have revised the manuscript**’s ending notes in the discussion, describing now quite specifically how our approach can be extended to larger problems and deeper networks. Namely, convolutional networks can be used, where SoftHebb is also applicable, very similarly to the fully connected case. More specifically, in the convolutional case, the WTA competition will be between convolutional kernels. This WTA computation and the corresponding SoftHebb update will be repeated over each patch of the layer’s input features, and this repeated operation over the patches can be parallelized on GPUs. We believe that this provides a very specific and realistic path to extending our approach.
> _edit: we have now performed proof of concept experiments with a convolutional version of SoftHebb, showing that it is functional and is applicable to more complex datasets such as CIFAR-10. Please see separate comment: https://openreview.net/forum?id=IJ-88dRfkdz&noteId=mYtjlaDQC3z_
>
> In addition, **we have performed new experiments** to address the reviewer’s criticism of our performance on F-MNIST. Specifically, we were previously reporting the accuracy of a network that was optimized for standard MNIST, which caused a low accuracy on F-MNIST. This choice aimed to show the relative generality of the approach without much optimization, but we understand that this point is overwhelmed by the low accuracy. We have now optimized the network for F-MNIST. SoftHebb reaches an accuracy of 87.46%, whereas the backpropagation-trained MLP reaches 90.55%. We have updated the manuscript’s text to reflect this, and we will also update the F-MNIST-related figures on adversarial robustness.
>
> _continued in next comment..._

---

> > ### Author Response · Authors · 2021-11-20
> > **Response to Reviewer ereX (part 2)**
> >
> > _...continued from previous comment_
> >
> > > comparison is need with other adversarial learning/robustness techniques
> >
> > **We did not present a new adversarial training technique, but rather a learning rule.** Therefore, we believe that **the relevant comparison is against other learning algorithms**, not against adversarial training or other ad hoc defence mechanisms. The novel insight that a biologically-plausible learning algorithm is more robust than backpropagation, specifically in the absence of adversarial defences, is quite intriguing, and its publication will trigger further research by the community. This first robustness result already in its current form, as well as our demonstrations of fast learning, are two important new entries to the list of benefits from such biologically-plausible machine learning, and also advance our understanding of biological intelligence.
> >
> > **We believe that we have addressed all issues raised by the reviewer.**
> >
> > **We suggest that the present surprising results will attract strong interest from many sub-communities of ICLR.**
> >
> > _continued in next comment... https://openreview.net/forum?id=IJ-88dRfkdz&noteId=mYtjlaDQC3z_

---

> ### Author Response · Authors · 2021-11-23
> **Response (part 3) New experiments: CIFAR-10 and convolutions**
>
> _...continued from previous comment_
>
> **We have performed additional experiments, testing the algorithm on larger problems and implementing more complex architectures.**
>
> We now report the network’s accuracy on **CIFAR-10** (49.78%).
>
> In addition, **we have implemented a convolutional version** of SoftHebb, and our tests show that it is able to learn. It achieves in fact a much improved accuracy of 98.63% on MNIST and 60.30% on CIFAR-10.
>
> Moreover, we have revised the manuscript to include **a new section** titled “Extensibility of SoftHebb: F-MNIST, CIFAR-10, conv-SoftHebb” (Section 3.4).
>
> We believe that, together with our earlier comments and changes to the manuscript, **we have now fully addressed all of the reviewer’s concerns.**

---

> ### Author Response · Authors · 2021-11-29
> **Concluding summary of the discussion with Reviewer ereX**
>
> Unfortunately, there has been no reaction from the Reviewer even though we have addressed all their comments. For clarity and posterity, we are summarising this discussion here.
>
> ---
> 1. > equivalency between mixture model and WTA has been proposed in a previous work (Moraitis et al. 2020)
>
> **The novel theoretical result is not that. Our main results are (a) that such an ANN/probabilistic model can be optimized by a Hebbian plasticity rule, and (b) the derivation of the specific rule.** This is new, it was a key missing step for the normative design of larger future Hebbian learning models, the derivation is not trivial at all, and this theory already led to the study and demonstration of novel practical advantages of speed and robustness.
>
> ---
> 2. > Will any of [other Hebbian and non-Hebbian learning rules] be adoptable in this framework?
>
> **Other rules do not provably optimize the same Bayesian model, or cannot do so without labels.** This further underlines the significance of our theoretical work.
>
> ---
> 3. > It is not clear what the advantage of softHebb is.
>
> **SoftHebb has multiple advantages that are detailed in the revised paper's introduction and discussion.** Summary:
> In the studied conditions, SoftHebb
> - solves simultaneously four inefficiencies and bio-implausibilities of backpropagation that have been under heavy research:
> -- non-locality
> -- update-locking
> -- weight-transport
> -- detailed feedback-dependence
> - is normative and backed by rigorous theory, contrary to many approaches to Hebbian learning that are heuristic
> - contrary to the standards of the biologically-plausible learning literature, SoftHebb even outperforms backpropagation in terms of accuracy in certain conditions
> - is surprisingly and significantly robust to noise and to adversarial attacks *without any defence mechanism*.
>
> ---
> 4. > poor on the F-MNIST dataset
>
> **We have updated the F-MNIST results**, now showing much smaller accuracy gap compared to backpropagation, while maintaining significant robustness advantages.
>
> ---
> 5. > Discussion and experiments on learning with larger/more complex datasets need to be addressed.
>
> **We have now provided results on CIFAR-10 and with convolutional SoftHebb, and added a section dedicated to extensibility**
>
> ---
> 6. > a comparison is need with other adversarial learning/robustness techniques
>
> **SofHebb does not use any such techniques, so it is only fair to compare it to other learning without such techniques, as we do.** SoftHebb's robustness is emergent, not ad-hoc.
>
> ---
>
> _**It is unfortunate that the Reviewer did not have a chance to react in time to our direct and timely rebuttals. We have fully addressed all points, and therefore we believe that the paper is ready for publication.**_ We do appreciate the initial feedback, which we fully used and addressed.

---

### Author Response · Authors · 2021-11-17
**Important context for the benefits of our approach (part 1)**

We thank the reviewers for their comments. Here we add important context that we believe sufficiently addresses their most high-level concerns.

**Our algorithm radically improves the backpropagation-based state of the art (SOTA). It does so on important fronts other than accuracy. It also shows advantages that were previously unknown in the Hebbian SOTA.**

SoftHebb’s central practical aim is not to surpass accuracy but to solve several other known inefficiencies of backpropagation. Several communities, especially those of computational neuroscience, bio-inspired machine learning, and neuromorphic computing, are more interested in improving SOTA in these terms, rather than improving accuracies.

In our experiments, the algorithm does manage to solve the following problems and limitations of backprop SOTA:
1.	Weight transport: there is no need for exactly symmetric forward-backward weights as in backpropagation, because there are no backward weights or backward propagation. This significantly simplifies circuit design in a potential hardware implementation
2.	Supervisory and non-local signals: in a potential implementation on a neuromorphic chip, there is no need for labels, or other feedback currents propagating throughout the chip, which consume energy
3.	Update-locking: in backpropagation, the weight update of a layer and the receipt of the subsequent input must wait for the forward pass to finish and the backward pass to arrive. In contrast to that, a SoftHebb layer is updated immediately without that locking.

These are in fact very important advances towards future models of neural computation that are the goal of large communities of ML research. Certain partly-biologically-plausible alternatives to backpropagation solve some of these problems, but, expectably, also trade off accuracy. Our present algorithm solves all of them, indicating the radical difference in the approach, so of course it also has to trade something off at this stage. Our advances could enable learning small tasks like our demonstrations, in edge devices with limited resources that cannot afford inefficiencies. On these efficiency fronts, our algorithm is a significant improvement of the backpropagation-based SOTA on these small tasks.

In the model category that solves these issues, namely competitive Hebbian learning, our algorithm is of course not the first one. However, despite the long literature on Hebbian learning, our work is the first of this kind to also surpass backpropagation accuracy, speed, and robustness on certain tasks, even if only marginally. That is while also having a rigorous theoretical description and a normative derivation of the model, and still solving the 3 important goals mentioned above. These together, we believe, are at least as significant and as non-trivial progress, as a strict focus on high accuracies in large problems and multilayer networks, and are expected to interest many large sub-communities of ICLR.

_continued in next comment..._

---

> ### Author Response · Authors · 2021-11-17
> **Important context for the benefits of our approach (part 2)**
>
> _...continued from previous comment_
>
> **Our focus on theory and shallow networks is a fundamental and non-trivial missing step before well-performing deep Hebbian networks.**
>
> We too look forward to presenting SoftHebb’s power on larger problems with multilayer implementations in a next paper. However, directly focusing on multilayer networks would not have been as productive. A number of previous Hebbian examples of multilayer networks do exist, however they have been lacking significantly in terms of accuracy compared to backpropagation. Other potential benefits such as robustness or speed were not known either. We believe that this is partly due to an incomplete understanding of each individual layer in a network, and an only partly normative guidance to each layer’s construction. This is the gap that our work fills.
>
> It was previously not known which precise plasticity rule to use, and in combination with which precise mechanism of competition within the layer, in order to learn to perform Bayesian computation in a Hebbian ANN layer. It was previously not known that a Hebbian network can have these robustness or learning-speed properties. These gaps are filled here by studying the shallow network on small problems, but, more fundamentally, by our rigorous theory that led to these experiments, and enabled these results.
> We believe that this previously-missing step is important for a solid path to deep Hebbian networks. This step is only possible through a first focus on shallow networks and thus small-scale experiments. Rejecting our paper will leave these gaps open.
>
> In the same time, it is clear that an additional section with multilayer experiments does not fit in the paper’s limits, if the necessary focus on single-layer theory and understanding is to be maintained. Essentially, presenting results from SoftHebb with multilayer networks might have produced good accuracies on more interesting problems, but at the expense of the more important insights that we provide here.
> _update: we have now included additional results on more complex datasets, and on a convolutional implementation of SoftHebb._
>
> We acknowledge that these results are not ground-breaking, but they are an important, necessary, and significantly non-trivial step in the literature’s timeline, including surprising results that we believe interest enough ICLR sub-communities to warrant publication.
>
> **Over the following days, we will also provide point-to-point responses to each reviewer, addressing all remaining details. However, we believe our comments here already warrant a re-evaluation of our work’s merit.**

---

### Author Response · Authors · 2021-11-23
**Summary of our updates**

Dear Reviewers,

Thank you for the constructive comments. We believe that **we have now addressed them all fully.**

**Our changes:**

We have:
- re-written parts of the manuscript to clarify the important benefits of our approach in the SOTA of alternatives to backpropagation, for readers that are not strictly in our specific field.
- performed a series of additional experiments on more complex data, with new architectures, comparing to different unsupervised models, and providing control experiments.
- taken all suggestions by the reviewers.
- clarified the novelty, significance, and non-triviality of our theoretical section.

We include here the revised Discussion section of our manuscript. Having addressed all weaknesses mentioned by the reviewers, we include it to also remind **the strengths of our work:**

>We have described SoftHebb, a highly biologically plausible neural algorithm that is founded on a Bayesian ML-theoretic framework. The model consists of elements fully compatible with conventional ANNs. It was previously not known which plasticity rule should be used to learn a Bayesian generative model of the input distribution in ANN WTA networks. Moreover, we showed that Hard WTA networks and neurons with other activation functions can be described within the same framework as variations of the probabilistic model. This theory could provide a new foundation for normative Hebbian ANN designs with practical significance. For example, SoftHebb's properties are sought-after by efficient neuromorphic learning chips. It is unsupervised, local, and requires no error or other feedback currents from upper layers, thus solving hardware-inefficiencies and bio-implausibilities of backpropagation such as weight-transport and update-locking. Surprisingly, it surpasses backpropagation even in accuracy, when training time and network size are limited. In a demonstration that goes beyond the common greedy-training approach to such networks, we achieved update-unlocked operation in practice, by updating the first layer before the input's full processing by the next layer. It is intriguing that, through its biological plausibility, emerge properties commonly associated with biological intelligence, such as speed of learning, and robustness to noise and adversarial attacks. Significant robustness emerges without specialized defences. Furthermore, SoftHebb tends to not merely be robust to attacks, but actually deflect them as specialized SOTA defences aim to do.

>Here, we explored SoftHebb's applicability on several datasets. We measured its accuracy on MNIST, Fashion-MNIST, and CIFAR-10 in preliminary results, and we reported a functional convolutional SoftHebb network that improves accuracy on the significantly harder dataset of CIFAR-10. The convolutional implementation could become the foundation for deeper networks and complex problems. Ultimately, this could provide insights into the role of WTA microcircuits in larger networks in cortex with localized receptive fields (Pogodin et al. 2021), similar to area V1 of cortex (Hubel and Wiesel, 1962).

>All in all, the algorithm has several properties that are individually interesting and novel, and worth future extension.
	Combined, however, SoftHebb's properties shown in this work may already enable certain small-scale but previously-impossible applications. For example, fast, on-line, unsupervised learning of simple tasks by edge sensing devices, operating in noisy conditions, with a small battery and only local processing, requires those algorithmic properties that we demonstrated here.

**We believe that the manuscript is now ready for publication. It contains important theoretical and intriguing experimental results that will benefit several large communities of ICLR .**

---

### Author Response · Authors · 2021-11-27
**Looking forward to your updated feedback**

Dear Reviewer ereX and Reviewer vMDj,

We have significantly updated the manuscript, including with results that specifically address your earlier criticisms. We have described the updates and the significance of the revised work in extensive comments here.

As the discussion period is due to close soon, we would certainly appreciate reading your updated feedback.

---

### Decision · Program_Chairs · 2022-01-20

**Decision:**

Reject

**Comment:**

The authors provide an analysis of soft-winner-take-all (WTA) networks with Hebbian local learning as a generative probabilistic mixture model. They then present experiments on comparably simple data sets, MNIST and F-MNIST. Results are compared to hard WTA networks and an MLP of the same size (single hidden layer) trained with backprop. Besides accuracy, the learning speed and adversarial robustness of the networks are compared.

This paper is borderline, and I was discussing it quite a bit with the reviewers.
The reviewers agree that the manuscript has some merits, but they also point to a number of weak points.

Besides the objective evaluation, I would like to comment on the review dynamics of this paper. The paper had initially rather low ratings. The authors commented extensively on the reviews, in several waves, and with suggestive text such as "All reviewer's points addressed" (as a comment title) or "Based on the Reviewer's earlier comment, the revised paper is now a clear 'accept'." to name just a few. I and the reviewers had the impression that the authors strongly urged the reviewers to increase their scores.

Due to the borderline ratings, I decided to read the paper carefully. My impression is in-line with the main criticisms of the reviewers, and summarized in the following:
On the positive side:
- The manuscript tackles an interesting problem. WTA architectures are biologically highly relevant structures and it is relevant to study learning in them.
- The authors provide a nice theoretical analysis.
- The observation that WTA architectures improve adversarial robustness is very interesting.
- Learning is local.
- The manuscript is well-written.

On the negative side:
- Theory: Similar analyses have been performed before. While there are differences, the main ideas are rather similar, in particular with respect to (Nessler et al., 2009). The authors argue that in contrast to their work, Nessler et al. 2009 deals with spiking neurons. But since the authors argue with biological plausibility, I would see that as an advantage of Nessler et al.
- Performance: The performance of their model is comparable to the standard hard WTA network, often showing only a very slight advantage. This raises the question why the soft WTA should be preferred over the hard WTA. The performance of the single-hidden-layer ANN is clearly better. This raises the question of the scalability of the approach.
- The analysis of adversarial robustness is interesting, but there is no comparison to other defense methods (e.g. adversarial training). The authors argued in their comments that it is not an adv. defense paper, so this comparison is out of scope. This reasoning is understandable, but since this is maybe the most interesting point of the paper, it would be a nice to have.
- Scalability: It is true that the learning is local, but the question is whether it scales to larger problems and deeper networks. After the first reviews, the authors added experiments on CIFAR-10 and a convolutional version of the model. However, the results were clearly below the state-of-the-art and the convolutional model is barely described (5 lines in the appendix).

Conclusion: The manuscript has some interesting points. Given the the strong competition within ICLR however, I cannot propose acceptance.